# High-throughput CRISPRi phenotyping identifies new essential genes in *Streptococcus pneumoniae*

Xue Liu[1,2] iD, Clement Gallay[1] iD, Morten Kjos[1,3] iD, Arnau Domenech[1] iD, Jelle Slager[1] iD, Sebastiaan P van Kessel[1], Kèvin Knoops[4], Robin A Sorg[1], Jing-Ren Zhang[2] iD & Jan-Willem Veening[1,5,*] iD

## Abstract

Genome-wide screens have discovered a large set of essential genes in the opportunistic human pathogen *Streptococcus pneumoniae*. However, the functions of many essential genes are still unknown, hampering vaccine development and drug discovery. Based on results from transposon sequencing (Tn-seq), we refined the list of essential genes in *S. pneumoniae* serotype 2 strain D39. Next, we created a knockdown library targeting 348 potentially essential genes by CRISPR interference (CRISPRi) and show a growth phenotype for 254 of them (73%). Using high-content microscopy screening, we searched for essential genes of unknown function with clear phenotypes in cell morphology upon CRISPRi-based depletion. We show that SPD_1416 and SPD_1417 (renamed to MurT and GatD, respectively) are essential for peptidoglycan synthesis, and that SPD_1198 and SPD_1197 (renamed to TarP and TarQ, respectively) are responsible for the polymerization of teichoic acid (TA) precursors. This knowledge enabled us to reconstruct the unique pneumococcal TA biosynthetic pathway. CRISPRi was also employed to unravel the role of the essential Clp-proteolytic system in regulation of competence development, and we show that ClpX is the essential ATPase responsible for ClpP-dependent repression of competence. The CRISPRi library provides a valuable tool for characterization of pneumococcal genes and pathways and revealed several promising antibiotic targets.

**Keywords** bacterial cell wall; competence; DNA replication; gene essentiality; teichoic acid biosynthesis
**Subject Categories** Chromatin, Epigenetics, Genomics & Functional Genomics; Genome-Scale & Integrative Biology; Microbiology, Virology & Host Pathogen Interaction
**Mol Syst Biol.** (2017) 13: 931

## Introduction

*Streptococcus pneumoniae* (pneumococcus) is a major cause of community-acquired pneumonia, meningitis, and acute otitis media and, despite the introduction of several vaccines, remains one of the leading bacterial causes of mortality worldwide (Prina *et al*, 2015). The main antibiotics used to treat pneumococcal infections belong to the beta-lactam class, such as amino-penicillins (amoxicillin, ampicillin) and cephalosporines (cefotaxime). These antibiotics target the penicillin binding proteins (PBPs), which are responsible for the synthesis of peptidoglycan (PG) that plays a role in the maintenance of cell integrity, cell division, and anchoring of surface proteins (Sham *et al*, 2012; Kocaoglu *et al*, 2015). The pneumococcal cell wall furthermore consists of teichoic acids (TA), which are anionic glycopolymers that are either anchored to the membrane (lipo TA) or covalently attached to PG (wall TA) and are essential for maintaining cell shape (Brown *et al*, 2013; Massidda *et al*, 2013). Unfortunately, resistance to most beta-lactam antibiotics remains alarmingly high. For example, penicillin non-susceptible pneumococcal strains colonizing the nasopharynx of children remain above 40% in the United States (Kaur *et al*, 2016), despite the effect of the pneumococcal conjugate vaccines. Furthermore, multidrug resistance in *S. pneumoniae* is prevalent and antibiotic resistance determinants and virulence factors can readily transfer between strains via competence-dependent horizontal gene transfer (Chewapreecha *et al*, 2014; Johnston *et al*, 2014; Kim *et al*, 2016). For these reasons, it is crucial to understand how competence is regulated and to identify and characterize new essential genes and pathways. Interestingly, not all proteins within the pneumococcal PG and TA biosynthesis pathways are known (Massidda *et al*, 2013), leaving room for discovery of new potential antibiotic targets. For instance, not all enzymes in the biosynthetic route to lipid II, the precursor of PG, are known and annotated in *S. pneumoniae*. The pneumococcal TA biosynthetic pathway is even more enigmatic, and it is unknown which genes code for the enzymes responsible for polymerizing TA precursors (Denapaite *et al*, 2012).

Several studies using targeted gene knockout and depletion/overexpression techniques as well as transposon sequencing (Tn-seq)

1 Molecular Genetics Group, Groningen Biomolecular Sciences and Biotechnology Institute, Centre for Synthetic Biology, University of Groningen, Groningen, The Netherlands
2 Center for Infectious Disease Research, School of Medicine, Tsinghua University, Beijing, China
3 Department of Chemistry, Biotechnology and Food Science, Norwegian University of Life Science, Ås, Norway
4 Molecular Cell Biology, Groningen Biomolecular Sciences and Biotechnology Institute, University of Groningen, Groningen, The Netherlands
5 Department of Fundamental Microbiology, Faculty of Biology and Medicine, University of Lausanne, Lausanne, Switzerland
*Corresponding author. Tel: +41 21 6925625; E-mail: Jan-Willem.Veening@unil.ch

have aimed to identify the core pneumococcal genome (Thanassi *et al*, 2002; Song *et al*, 2005; van Opijnen *et al*, 2009; van Opijnen & Camilli, 2012; Zomer *et al*, 2012; Mobegi *et al*, 2014; Verhagen *et al*, 2014). These genome-wide studies revealed a core genome of around 400 genes essential for growth either *in vitro* or *in vivo*. Most of the essential pneumococcal genes can be assigned to a functional category on the basis of sequence homology or experimental evidence. However, per the most recent gene annotation of the commonly used *S. pneumoniae* strain D39 (NCBI, CP000410.1, updated on 31-JAN-2015), approximately one-third of the essential genes belong to the category of "function unknown" or "hypothetical" and it is likely that several unknown cell wall synthesis genes, such as the TA polymerase, are present within this category.

To facilitate the high-throughput study of essential genes in *S. pneumoniae* on a genome-wide scale, we established CRISPRi (clustered regularly interspaced short palindromic repeats interference) for this organism. CRISPRi is based on expression of a nuclease-inactive *Streptococcus pyogenes* Cas9 (dCas9), which together with expression of a single-guide RNA (sgRNA) targets the gene of interest (Bikard *et al*, 2013; Qi *et al*, 2013; Peters *et al*, 2016). When targeting the non-template strand of a gene by complementary base-pairing of the sgRNA with the target DNA, the dCas9-sgRNA-DNA complex acts as a roadblock for RNA polymerase (RNAP) and thereby represses transcription of the target genes (Qi *et al*, 2013; Peters *et al*, 2016) (Fig 1A). Note that *S. pneumoniae* does not contain an endogenous CRISPR/Cas system, consistent with interference with natural transformation and thereby lateral gene transfer that is crucial for pneumococcal host adaptation (Bikard *et al*, 2012).

Using Tn-seq and CRISPRi, we refined the list of genes that are either essential for viability or for fitness in *S. pneumoniae* strain D39 (Avery *et al*, 1944). To identify new genes involved in pneumococcal cell envelope homeostasis, we screened for essential genes of unknown function (as annotated in NCBI), with a clear morphological defect upon CRISPRi-based depletion. This identified SPD_1416 and SPD_1417 as essential peptidoglycan synthesis proteins (renamed to MurT and GatD, respectively) and SPD_1198 and SPD_1197 as essential proteins responsible for precursor polymerization in TA biosynthesis (hereafter called TarP and TarQ, respectively). Finally, we demonstrate the use of CRISPRi to unravel gene regulatory networks and show that ClpX is the ATPase subunit that acts together with the ClpP protease as a repressor for competence development.

# Results

## Identification of potentially essential genes in *S. pneumoniae* strain D39

While several previous studies have identified many pneumococcal genes that are likely to be essential, the precise contribution to pneumococcal biology has remained to be defined for most of these genes. Here, we aim to characterize the functions of these proteins in the commonly used *S. pneumoniae* serotype 2 strain D39 by the CRISPRi approach. Therefore, we performed Tn-seq on *S. pneumoniae* D39 grown in C+Y medium at 37°C, our standard laboratory

condition (see Materials and Methods). We included all genes that we found to be essential in our Tn-seq study, and added extra genes that were found to be essential by previous Tn-seq studies with a serotype 4 strain TIGR4 (van Opijnen *et al*, 2009; van Opijnen & Camilli, 2012) in the CRISPRi library (see below). Finally, 391 potentially essential genes were selected, and the genes are listed in Dataset EV1.

## CRISPRi enables tunable repression of gene transcription in *S. pneumoniae*

To develop the CRISPR interference system, we first engineered the commonly used LacI-based isopropyl β-D-1-thiogalactopyranoside (IPTG)-inducible system for *S. pneumoniae* (see Materials and Methods). The *dcas9* gene was placed under control of this new IPTG-inducible promoter, named P$_{lac}$, and was integrated into the chromosome via double crossover (Fig 1A and B). To confirm the reliability of the CRISPRi system, we tested it in a reporter strain expressing firefly luciferase (*luc*), in which an sgRNA targeting *luc* was placed under the constitutive P3 promoter (Sorg *et al*, 2015) and integrated at a non-essential locus (Fig 1B). To obtain high efficiency of transcriptional repression, we used the optimized sgRNA sequence as reported previously (Chen *et al*, 2013) (Fig EV1A).

Induction of dCas9 with 1 mM IPTG resulted in quick reduction in luciferase activity; ~30-fold repression of luciferase expression was obtained within 2 h without substantial impact on bacterial growth (Fig 1C). Furthermore, the level of repression was tunable by using different concentrations of IPTG (Fig 1C). To test the precision of CRISPRi in *S. pneumoniae*, we determined the transcriptome of the sgRNA*luc* strain (strain XL28) by RNA-Seq in the presence or absence of IPTG. The data were analyzed using Rockhopper (McClure *et al*, 2013) and T-REx (de Jong *et al*, 2015). The RNA-Seq data showed that the expression of dCas9 was stringently repressed by LacI without IPTG and was upregulated ~600-fold upon addition of 1 mM IPTG after 2.5 h. Upon dCas9 induction, the *luc* gene was significantly repressed (~84-fold) (Fig 1D). Our RNA-Seq data showed that the genes (*spd_0424*, *spd_0425*, *lacE-1*, *lacG-1*, *lacF-1*) that are downstream of *luc*, which was driven by a strong constitutive promoter without terminator, were significantly repressed as well (Appendix Fig S1A). This confirms the reported polar effect of CRISPRi (Qi *et al*, 2013). In addition, induction of dCas9 in the sgRNA-deficient strain XL29 (Fig EV1B) led to no repression of the target gene (Fig EV1C). By comparing strains with or without sgRNA*luc*, we found that repression in our CRISPRi system is stringently dependent on the expression of both dCas9 and the sgRNA, and detected no basal level repression (Fig EV1C). Furthermore, we compared the transcriptome of *luc* reporter strains with sgRNA*luc* (strain XL28) and without sgRNA*luc* (strain XL29) both grown in the presence of 1 mM IPTG. This showed that *galT-2*, *galK*, and *galR* were upregulated in both strains, indicating that these genes are activated in response to the inducer IPTG and not by the CRISPRi system itself (Dataset EV2). We also noted a slight repression of several competence genes in both XL28 and XL29 with 1 mM IPTG (Dataset EV2). Since this repression does not rely on the presence of a functional CRISPRi system, we anticipate that these changes are due to the noisy character of the competence system (Aprianto *et al*, 2016; Prudhomme *et al*, 2016). Taken together, the IPTG-inducible CRISPRi system is highly specific.

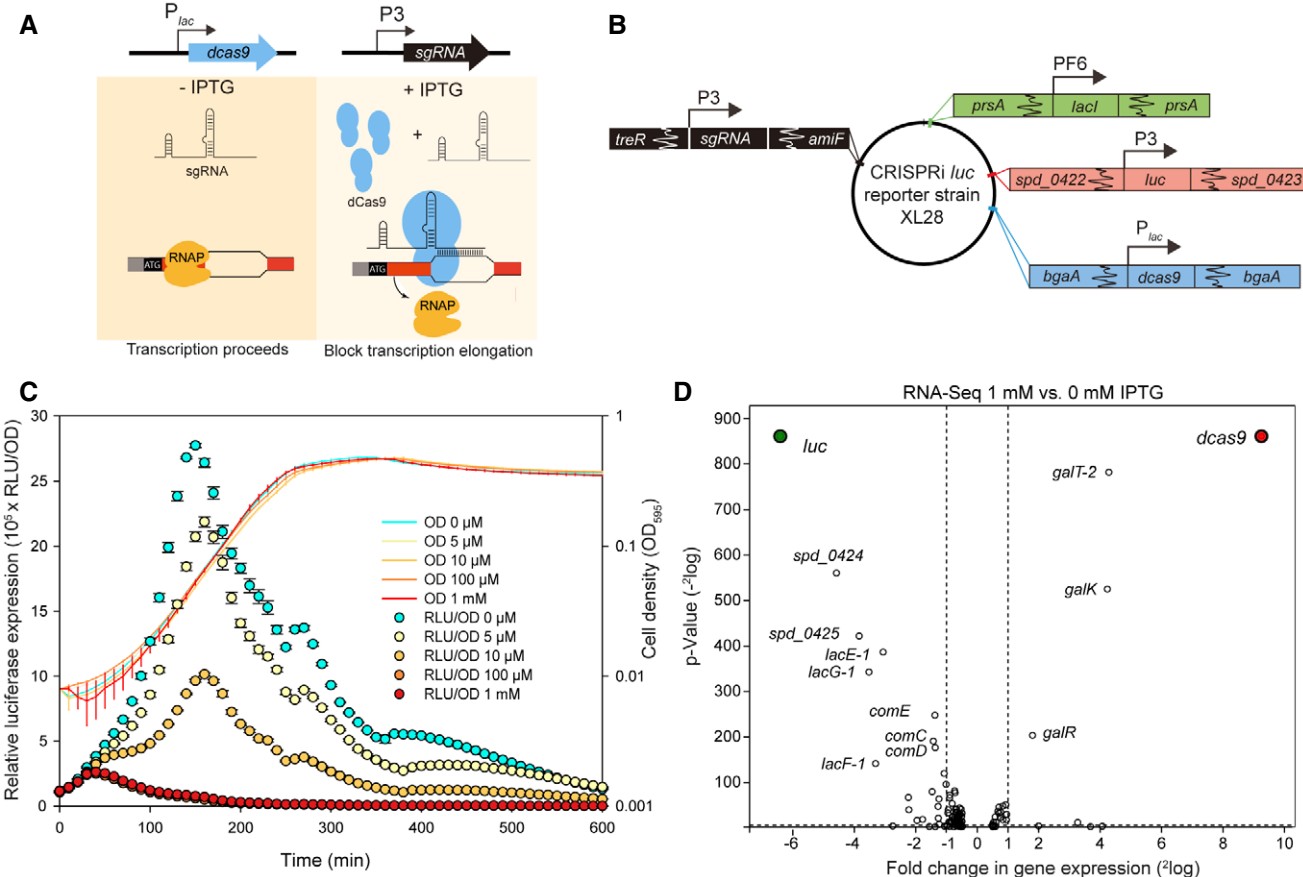

**Figure 1. An IPTG-inducible CRISPRi system for tunable repression of gene expression in *S. pneumoniae*.**

A   *dcas9* and sgRNA sequences were chromosomally integrated at two different loci, and expression was driven by an IPTG-inducible promoter ($P_{lac}$) and a constitutive promoter (P3), respectively. With addition of IPTG, dCas9 is expressed and guided to the target site by constitutively expressed sgRNA. Binding of dCas9 to the 5′ end of the coding sequence of its target gene blocks transcription elongation. In the absence of IPTG, expression of dCas9 is tightly repressed, and transcription of the target gene can proceed smoothly.

B   Genetic map of CRISPRi *luc* reporter strain XL28. To allow IPTG-inducible expression, the *lacI* gene, driven by the constitutive PF6 promoter, was inserted at the non-essential *prsA* locus; *luc*, encoding firefly luciferase, driven by the constitutive P3 promoter was inserted into the intergenic sequence between gene loci *spd_0422* and *spd_0423*; *dcas9* driven by the IPTG-inducible $P_{lac}$ promoter was inserted into the *bgaA* locus; sgRNA-*luc* driven by the constitutive P3 promoter was inserted into the CEP locus (between *treR* and *amiF*).

C   The CRISPRi system was tested in the *luc* reporter strain XL28. Expression of *dcas9* was induced by addition of different concentrations of IPTG. Cell density ($OD_{595}$) and luciferase activity (shown as RLU/OD) of the bacterial cultures were measured every 10 min. The values represent averages of three replicates with SEM.

D   RNA-Seq confirms the specificity of the CRISPRi system in *S. pneumoniae*. RNA sequencing was performed on the *luc* reporter strain XL28 (panel B) with or without 1 mM IPTG. The *dcas9* and *luc* genes are highlighted. Data were analyzed with T-REx and plotted as a volcano plot. *P*-value equals 0.05 is represented by the horizontal dotted line. Two vertical dotted lines mark the twofold changes.

## Construction and growth analysis of the CRISPRi library

We next used the CRISPRi system to construct an expression knock-down library of pneumococcal essential genes. An sgRNA to each of the 391 potentially essential genes was designed as described previously (Larson *et al*, 2013) (Dataset EV3). Based on the sgRNA*luc* plasmid (Fig 2A), we tested two different cloning strategies to introduce the unique 20-nt base-pairing region for each gene: infusion cloning and inverse PCR (Ochman *et al*, 1988; Irwin *et al*, 2012; Larson *et al*, 2013) (Fig EV2A). For infusion cloning, we synthesized two complementary primers consisting of the 20-nt base-pairing region flanked by 15-nt overlap sequences. The two complementary primers were then annealed to form a duplex DNA fragment and cloned into the vector by the infusion reaction, followed by direct transformation into *S. pneumoniae* D39 strain DCI23. With inverse PCR, we used a phosphorylated universal primer, together with a gene-specific primer to fuse the 20-nt base-pairing region into the vector by PCR, followed by blunt-end ligation and direct transformation into *S. pneumoniae* D39 strain DCI23. We compared the efficiency of the two methods by creating sgRNA strains targeting the known essential gene *folA* (*spd_1401*). Depletion of *folA* causes a clear growth defect, which could thus be used to test the functionality of sgRNA*folA* in transformants. We found that 79% of the transformants produced by infusion cloning had a growth defect upon dCas9 induction with IPTG (38 out of 48 colonies), whereas 26% of the transformants generated by inverse PCR

showed a phenotype (12/46). Sequencing validated that transformants with a growth defect contained the correct sgRNA sequence. Considering the convenience and efficiency, we adopted the infusion cloning strategy for sgRNA cloning in this study. All sgRNA constructs were sequence verified, and we considered them genetically functional when the sgRNA did not contain more than 1 mismatch to the designed sgRNA and no mismatches in the first 14-nt prior to the PAM. Using this approach, after a single round of cloning and sequencing, we successfully constructed 348 unique sgRNA strains (see Materials and Methods). Note that we are still in the process of constructing the remaining 43 sgRNA strains, the failure of which is likely caused by technical reasons (e.g., incorrect oligonucleotides, poor oligo annealing, low transformation).

To examine the effects of CRISPRi-based gene silencing, growth was assayed both in the presence and absence of 1 mM IPTG for 18 h in real time by microtiter plate assays. Two types of growth

phenotypes were defined and identified: a growth defect and increased lysis (Fig EV2B–E). As shown in Fig 2B, CRISPRi-based repression of transcription led to a growth defect in 230 genes, 48 genes showed increased lysis, including 24 that demonstrated both a growth defect and increased lysis, and 94 genes showed no defect (see Dataset EV1). In total, 254 out of 348 target genes (about 73%) repressed by CRISPRi showed growth phenotypes. Comparing the optical densities between the uninduced and induced cells at the time point at which uninduced cells reached an $OD_{595}$ of ~0.1, 174 genes repressed by CRISPRi displayed a more than fourfold growth defect, and 254 genes showed a more than twofold growth defect (Fig 2C). To further validate the specificity of the CRISPRi system, CRISPRi strains targeting eight genes identified as essential and eight genes as dispensable by Tn-seq were included in the growth analysis. The selected dispensable genes are present as a monocistron or are in an operon with other

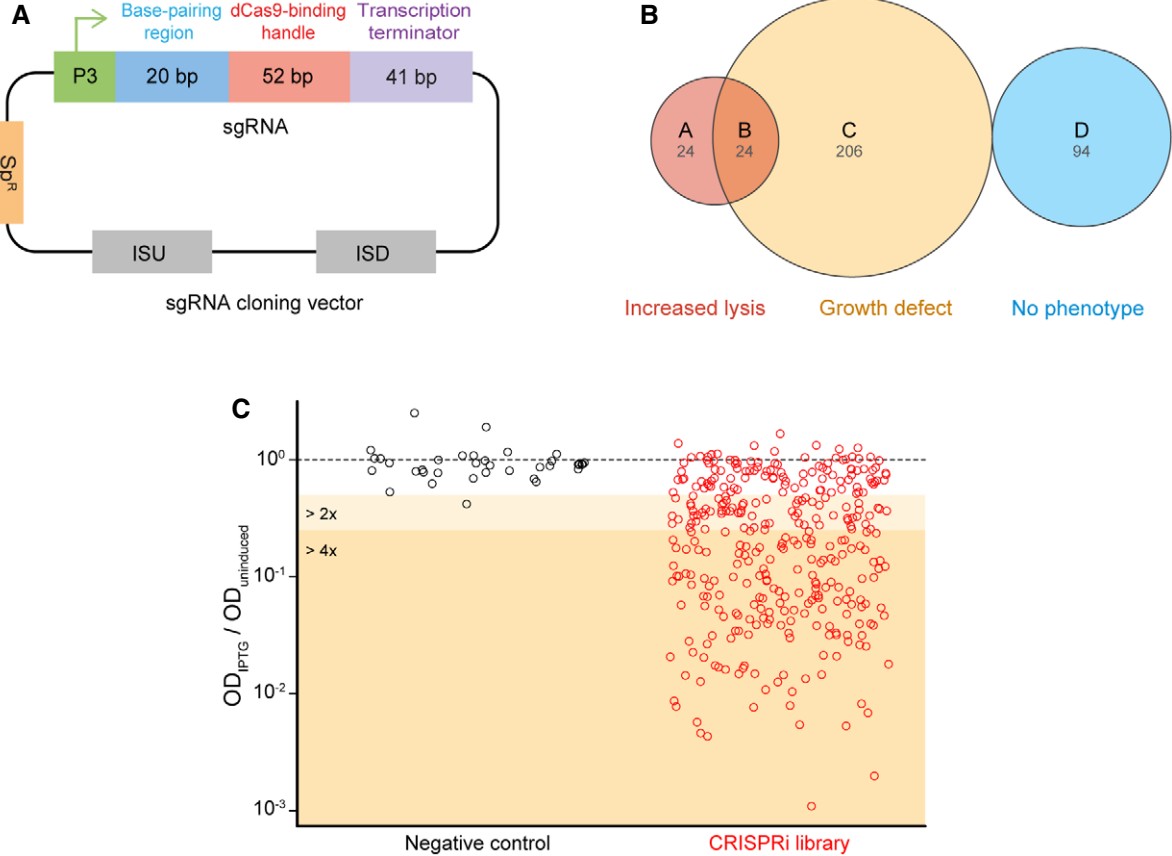

**Figure 2. Construction and growth analysis of the CRISPRi library.**

A       The plasmid map of the sgRNA cloning vector (pPEPX-P3-sgRNA*luc*). The sgRNA expression vector is a *S. pneumoniae* integration vector. It contains a constitutive P3 promoter, a spectinomycin-selectable marker ($Sp^R$), two homology sequences (ISU and ISD) for double crossover integration at the CEP locus (Sorg *et al*, 2015), and the sgRNA sequence. The sgRNA chimera contains a base-pairing region (blue), dCas9-binding handle (red), and the *S. pyogenes* transcription terminator (purple).

B, C   Growth analysis of the whole library. (B) Classification of the 348 genes targeted by the CRISPRi library according to growth analysis: A represents the 24 strains that only showed increased autolysis; B represents the 24 strains showing both increased autolysis and growth defects; C represents the 206 strains that showed only growth defects; D represents the 94 strains with no phenotype. Criteria for determination of a growth defect and increased lysis are demonstrated in Fig EV2B–E. (C) Comparison of the $OD_{595}$ of IPTG-induced cells ($OD_{IPTG}$) to the $OD_{595}$ of uninduced cells ($OD_{uninduced}$) at a time point. The time point at which uninduced cells have an optical density (595 nm) closest to 0.1 was selected for the plotting. *y*-axis represents the value of $OD_{IPTG}$ divided by $OD_{uninduced}$. The red data points in the dark orange area (174/348 strains) correspond to strains displaying a strong growth defect (more than fourfold); points in the light orange area demonstrate a moderate growth defect of twofold to fourfold (71/348 strains). The same type of analysis was performed on 36 negative control strains, shown as the black data points.

non-essential genes. As shown in Fig EV3A, no apparent growth defects could be observed when these non-essential genes were targeted by CRISPRi, while repression of essential genes led to strong growth defects (Fig EV3B).

It should be noted that CRISPRi repression of dispensable genes that are cotranscribed with essential genes can lead to growth phenotypes (Appendix Fig S1), which is due to polar effect of CRISPRi system (Qi *et al*, 2013). Thus, some of the genes may be targeted multiple times in the CRISPRi library (in case of more than one essential gene within the operon). We also observed that after a lag phase, most CRISPRi knockdowns with growth phenotypes eventually grow out to the same final OD (Fig EV4A). Re-culturing these cells showed the absence of sensitivity to IPTG, indicative of the presence of suppressor mutations (Fig EV4A). Indeed, by sequencing the two key elements of the CRISPRi system, the sgRNA and *dcas9*, we found that most of the suppressor strains contain loss-of-function mutation in the *dcas9* coding sequence (Fig EV4B). This is similar to observations made for the CRISPRi system in *Bacillus subtilis* (Zhao *et al*, 2016).

## Phenotyping pneumococcal genes by combined CRISPRi and high-content microscopy

To test whether CRISPRi was able to place genes in a functional category and thereby allow us to identify previously uncharacterized genes with a function in cell envelope homeostasis, we first analyzed the effects of CRISPRi-based repression on cell morphology using 68 genes. These genes were selected as they represent different functional pathways and have been identified as essential or crucial for normal pneumococcal growth by Tn-seq studies (van Opijnen *et al*, 2009; van Opijnen & Camilli, 2012) and by displaying strong growth phenotypes in our CRISPRi assay (Fig 2B and C). The selected genes have been associated with capsule synthesis (three genes), transcription (four genes), cell division (six genes), translation (seven genes), teichoic acid biosynthesis (nine genes), cell membrane synthesis (11 genes), chromosome biology (14 genes), and peptidoglycan synthesis (14 genes) (Table 1). High-content microscopy of the CRISPRi knockdowns showed a good correlation between reported gene function and observed phenotype. The common features of the morphological changes caused by CRISPRi repression of genes belonging to the same functional categories are summarized in Table 1. Growth analysis and microscopy phenotyping of a representative gene of each pathway, CRISPRi repression of which showed typical morphological changes of its pathway, were included in Fig 3. Morphological changes of CRISPRi repression of the other genes of the pathways are shown in Appendix Figs S2–S9. For instance, compared with the control strain (Fig 3A, XL28), repression of transcription of genes involved in chromosome biology caused, as expected, appearance of anucleate cells or cells with aberrant chromosomes (Fig 3B, *dnaA*; Appendix Fig S2). Cells with repression of genes involved in transcription showed a significant growth defect, and no obvious morphological changes were observed (Fig 3C, *rpoC*; Appendix Fig S3). Repression of genes involved in translation showed heterogeneous cell shapes and condensed nucleoids (Fig 3D, *infC*; Appendix Fig S4), in line with our previous observations (Sorg & Veening, 2015) and observations made in *Escherichia coli* showing that inhibition of protein synthesis by antibiotics leads to nucleoid

condensation (Morgan *et al*, 1967; Zusman *et al*, 1973; Roggiani & Goulian, 2015).

In *S. pneumoniae*, the fatty acid biosynthesis genes are all located in a single cluster (Lu & Rock, 2006) (Appendix Fig S5A), and two promoters in front of *fabT* and *fabK* are regulated by the transcriptional repressor FabT (Jerga & Rock, 2009). It was shown that *fabT* and *fabH* are cotranscribed (Lu & Rock, 2006), but the transcription pattern of the other genes is still unknown, which makes functional study of these genes with CRISPRi very difficult due to polar effects of the block of transcription elongation (Qi *et al*, 2013). Nevertheless, repression of transcription of genes involved in cell membrane synthesis caused diverse patterns of morphological changes: repression of *fabH*, *acpP*, *fabK*, *fabD,* and *fabG* led to a spotty Nile red pattern and irregular cell shapes including more pointy cells (Fig 3E, *fabK*; Appendix Fig S5B), as was shown previously (Kuipers *et al*, 2016); repression of *fabF*, *accB*, *fabZ*, and *accD* led to chaining of cells, heterogeneous cell sizes and irregular cell shapes; repression of *acpS* resulted in elongated and enlarged cells, whereas repression of *cdsA* caused cell rounding with heterogeneous cell sizes (Appendix Fig S5B).

When transcription of genes involved in cell division was repressed, we observed cells with irregular shapes and heterogeneous sizes (Appendix Fig S6). Interestingly, repression of *ftsZ* and *ftsL* caused similar morphological changes (Fig 3F, *ftsZ*; Appendix Fig S6), consistent with the reported function of FtsL on regulating FtsZ ring (Z-ring) dynamics in *B. subtilis* (Kawai & Ogasawara, 2006). Cells with repression of *ezrA* formed twisting chains and contained multiple septa, some of which formed at cell poles instead of midcell. Indeed, it was reported that *B. subtilis* EzrA can modulate the frequency and position of the Z-ring formation (Chung *et al*, 2004).

Repression of genes involved in capsule synthesis caused aggregation of cells (Appendix Fig S7), which may be due to the reduction in the negatively charged capsule that can provide a repelling electrostatic force preventing cell aggregation (Li *et al*, 2013).

Repression of transcription of genes involved in cell wall synthesis caused different phenotypes, depending on which step in peptidoglycan synthesis was interrupted. *S. pneumoniae* is oval-shaped, and it displays both septal and peripheral growth (Massidda *et al*, 2013; Pinho *et al*, 2013). Peptidoglycan synthesis of *S. pneumoniae* starts from formation of UDP-MurNAc-pentapeptides. Repression of expression of genes playing roles in these very first steps, including *glmU*, *alr*, *ddl*, *murI*, *murC*, *murD*, *murE*, and *murF*, will block both septal and peripheral peptidoglycan synthesis. Consistent with this prediction, we observed severe changes in cell shape and size, including heterogeneous cell sizes, exploding cells, defective septa, round cells, and cells demonstrating a coccus-to-rod transition (Appendix Fig S8). MraY and MurG play roles in formation of lipid II, and they are thus also involved in both peripheral and septal peptidoglycan synthesis. CRISPRi strains repressing *mraY* or *murG* led to a mix of elongated cells and short cells (Appendix Fig S8). FtsW and RodA are members of SEDS (shape, elongation, division, and sporulation) proteins (Meeske *et al*, 2016) and were first identified in *E. coli* (Ikeda *et al*, 1989). Inactivation of FtsW in *E. coli* blocks cell division without an effect on cell elongation (Khattar *et al*, 1994), and FtsW is suggested to act as a lipid II flippase (Mohammadi *et al*, 2011). FtsW of *S. pneumoniae* was believed to have a conserved function with *E. coli* (Maggi *et al*, 2008), is

**Table 1. Cellular pathways selected for CRISPRi phenotyping.**

| Pathway | Phenotype | Gene[a] | Pathway | Phenotype | Gene[a] |
|---|---|---|---|---|---|
| Chromosome replication | Anucleate cells; longer chains; uneven distribution of chromosomes; heterogeneous cell size | **dnaA (SPD_0001)** | Cell division | Exploding cells; heterogeneous cell size; defective septa; twisting chains | ftsL (SPD_0305) |
| | | dnaN (SPD_0002) | | | gpsB (SPD_0339) |
| | | gyrB (SPD_0709) | | | ftsE (SPD_0659) |
| | | parE (SPD_0746) | | | ftsX (SPD_0660) |
| | | parC (SPD_0748) | | | ezrA (SPD_0710) |
| | | dnaX (SPD_0760) | | | **ftsZ (SPD_1479)** |
| | | ftsK (SPD_0774) | Capsule synthesis | Cell aggregation; heterogeneous cell size | cps2E (SPD_0319) |
| | | dnaG (SPD_0957) | | | cps2I (SPD_0324) |
| | | xerS (SPD_1023) | | | cps2L (SPD_0328) |
| | | gyrA (SPD_1077) | Peptidoglycan biosynthesis | Heterogeneous cell size; coccus-to-rod transition; round cells; elongated cells; enlarged cells; defective septa | uppS (SPD_0243) |
| | | dnaI (SPD_1521) | | | **pbp2X (SPD_0306)** |
| | | priA (SPD_1546) | | | mraY (SPD_0307) |
| | | parB (SPD_2069) | | | uppP (SPD_0417) |
| | | dnaC (SPD_2030) | | | murD (SPD_0598) |
| Transcription | No strong morphological phenotype | rpoA (SPD_0218) | | | murG (SPD_0599) |
| | | rpoD (SPD_0958) | | | rodA (SPD_0706) |
| | | **rpoC (SPD_1758)** | | | glmU (SPD_0874) |
| | | rpoB (SPD_1759) | | | ftsW (SPD_0952) |
| Translation | Condensed nucleoids; short cells; heterogeneous cell size | rpsJ (SPD_0192) | | | murE (SPD_1359) |
| | | rplD (SPD_0194) | | | murF (SPD_1483) |
| | | rplV (SPD_0198) | | | ddl (SPD_1484) |
| | | rpsC (SPD_0199) | | | alr (SPD_1508) |
| | | efp (SPD_0395) | | | murI (SPD_1661) |
| | | **infC (SPD_0847)** | Teichoic acid biosynthesis | Longer chains; elongated cells; enlarged cells; heterogeneous cell size; defective septa | SPD_0099 |
| | | tsf (SPD_2041) | | | licC (SPD_1123) |
| Cell membrane biosynthesis | Spotty membrane staining; irregular cell shape; heterogeneous cell size | cdsA (SPD_0244) | | | licB (SPD_1124) |
| | | fabH (SPD_0380) | | | licA (SPD_1125) |
| | | acpP (SPD_0381) | | | tarJ (SPD_1126) |
| | | **fabK (SPD_0382)** | | | tarI (SPD_1127) |
| | | fabD (SPD_0383) | | | SPD_1200 |
| | | fabG (SPD_0384) | | | **licD3 (SPD_1201)** |
| | | fabF (SPD_0385) | | | SPD_1620 |
| | | accB (SPD_0386) | | | |
| | | fabZ (SPD_0387) | | | |
| | | accD (SPD_0389) | | | |
| | | acpS (SPD_1509) | | | |

[a]The genes highlighted in bold were included in Fig 3.

co-localized with septal HMW (high molecular weight) PBPs (Morlot *et al*, 2004), and is thus predicted to be involved in septal peptidoglycan synthesis. By morphological analysis, we provided experimental evidence to support this prediction: FtsW and Pbp2X are responsible for septal peptidoglycan synthesis, and elongated cells and coccus-to-rod transition were observed with CRISPRi repression of *ftsW* or *pbp2X* (Fig 3G, *pbp2X*; Appendix Fig S8, *ftsW*). RodA of *S. pneumoniae* shows 26% identity with RodA of

*E. coli* (Noirclerc-Savoye *et al*, 2003), which is required for cell elongation. Studies of RodA in *B. subtilis* also support its function on elongation of the lateral wall (Henriques *et al*, 1998; Meeske *et al*, 2016). RodA of *S. pneumoniae* was predicted to be a lipid II flippase responsible for peripheral peptidoglycan synthesis (Massidda *et al*, 2013). *Streptococcus pneumoniae* cells with repressed *rodA* expression by CRISPRi are consistently shorter (Appendix Fig S8), indicating a defect in cell elongation.

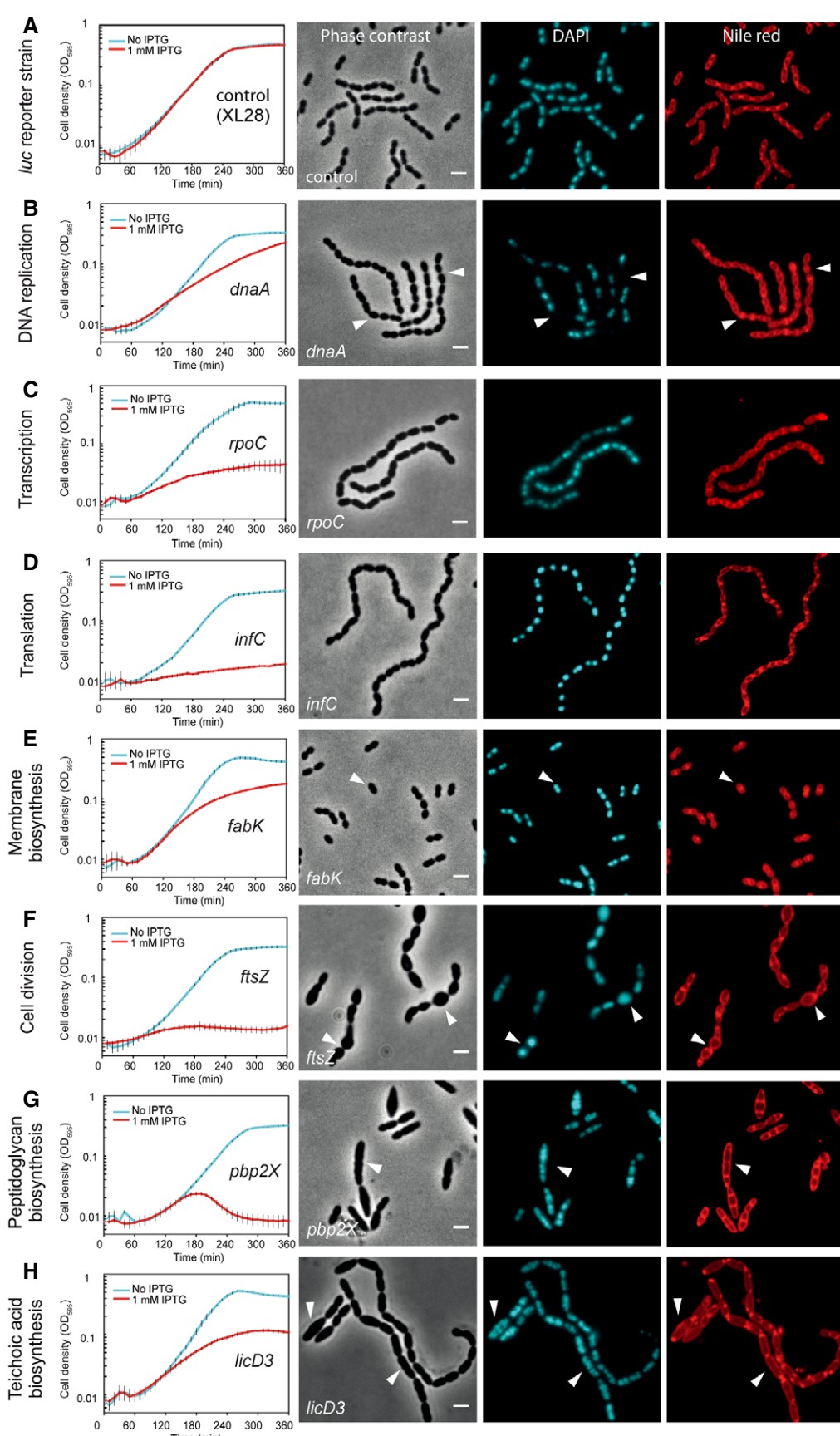

**Figure 3.**

**Figure 3. Growth profiles and morphological changes of CRISPRi strains with sgRNA targeting genes of different functional pathways.**

A–H    Growth of *S. pneumoniae* strains was performed in C+Y medium with (red) or without (cyan) 1 mM IPTG. Cell densities were measured every 10 min. The values represent averages of three replicates with SEM. Morphological changes were examined with fluorescence microscopy, and representative micrographs are shown. Phase contrast, DAPI staining, and Nile red staining are displayed. Scale bars = 2 μm. *Streptococcus pneumoniae* D39 reporter strain XL28 expresses firefly luciferase (*luc*) from a constitutive promoter and contains an sgRNA targeting the *luc* gene, and serves as a control strain without growth defects. White arrows point to specific (morphological) changes. For *dnaA*, arrows point to anucleate cells (B); *fabK*, non-uniform, spotty membrane staining (E); *ftsZ*, ballooning cells (F); *pbp2X*, elongated cells (G); *licD3*, elongated and enlarged cells (H). Repression of a transcription-related gene, *rpoC*, no strong morphological changes were observed (C); a translation-related gene, *infC*, led to generally condensed chromosomes as shown in the DAPI staining image (D). One gene of each pathway was presented in this figure. Additional information related to this figure can be found in Table 1 and Appendix Figs S2–S9, which show microscopy images of more genes of each pathway.

Repression of genes involved in teichoic acid (TA) biosynthesis led to morphological changes, including formation of longer chains and cells of heterogeneous sizes, mostly enlarged or elongated (Fig 3H, *licD3*; Appendix Fig S9). Growth of *S. pneumoniae* depends on exogenous choline, which is an essential molecule for the synthesis of pneumococcal TA, and the chaining phenotype caused by repression of genes involved in TA synthesis is in line with *S. pneumoniae* growing in medium without choline (Damjanovic *et al*, 2007).

In summary, by morphological analysis of CRISPRi strains for repression of transcription of genes with known function from different pathways, we established links between genotypes and phenotypes. Importantly, repression of transcription of genes known to be involved in cell envelope homeostasis, such as *ftsZ*, *ftsL*, *ftsW*, *rodA*, *pbp2X*, *glmU*, *murC*, *murF*, *tarI*, *tarJ*, *licA*, *licB*, *licC*, and *licD3*, caused severe changes in cell shape and size, including heterogeneous cell size, ballooning cells, defective septa, short cells, round cells, cells in chains, and cells demonstrating a coccus-to-rod transition. These observations provide a useful platform for the functional identification of hypothetical genes, especially genes involved in cell envelope homeostasis.

### Functional verification and annotation of *pcsB* (*spd_2043*), *vicR* (*spd_1085*), *divIC* (*spd_0008*), and *rafX* (*spd_1672*)

We next analyzed 44 strains in the CRISPRi library that target genes that are annotated as hypothetical in the *S. pneumoniae* D39 genome in the NCBI database (CP000410.1, updated on 31-JAN-2015). From this approach, we were able to verify the function and annotate several genes, whose function had been studied in pneumococci before but have not been properly annotated in the D39 genome. For example, repression of genes (*spd_0008*, *spd_1085*, and *spd_2043*) led to significant growth defects and cell shape and cell size changes (Appendix Fig S11A and B). Knocking down *spd_2043* and *spd_1085* led to almost the same morphological changes, which included irregular cell shape, heterogeneous cell sizes, and appearance of ballooned cells, suggesting that these two genes might be functionally associated and play roles in peptidoglycan synthesis or cell division. By literature mining and BLAST searches, we recognized *spd_1085* as *vicR* and *spd_2043* as *pcsB* (Ng *et al*, 2003). Consistent with the observed phenotypes in the CRISPRi strains, *pcsB* was shown to be essential for cell wall separation and its expression relies on the response regulator encoded by *vicR* (Reinscheid *et al*, 2001; Ng *et al*, 2003; Sham *et al*, 2011; Bartual *et al*, 2014). Similarly, the morphological changes suggested a potential role of SPD_0008 in cell wall synthesis or cell division. In line with this, SPD_0008 was identified as DivIC, which was reported to form a trimeric complex with DivIB and FtsL and

colocalized at division sites of *S. pneumoniae* strain R6 (Noirclerc-Savoye *et al*, 2005).

CRISPRi knockdown strain targeting *spd_1672* showed no significant growth defect at exponential phase, but cells lysed quicker in the stationary phase (Appendix Fig S11C). Microscopy showed that bacterial cells with CRISPRi-repressed *spd_1672* formed significantly longer chains (Appendix Fig S11D). Chained cells displayed irregular shapes and heterogeneous cell sizes. These phenotypes are very similar to the morphological changes caused by repression of genes involved in the biosynthesis of teichoic acid (Appendix Fig S9). Actually, *spd_1672* has been studied in *S. pneumoniae* R6 and was shown to contribute to the biosynthesis of wall teichoic acid and was named *rafX* (Wu *et al*, 2014). The reported *spd_1672* knockout strain of *S. pneumoniae* R6 also displayed a reduced stationary phase with similar cell shape and cell size defects. Inconsistent with our study, longer chains were not observed by TEM (transmission electron microscopy) imaging in the Wu *et al* study. To exclude the possible polar effect of CRISPRi repression, we made a *spd_1672* knockout in *S. pneumoniae* D39, and the *spd_1672* knockout strain also showed longer chains. Thus, the mismatch in phenotypes between the studies may be due to the different genetic background of *S. pneumoniae* D39 and R6, or may be caused by the process of sample preparation for TEM examination.

### Annotation and characterization of chromosome replication genes *dnaB* (*spd_1522*), *dnaD* (*spd_1405*), and *yabA* (*spd_0827*)

High-content microscopy screening of the CRISPRi library showed that repression of *spd_1405*, *spd_1522*, and *spd_0827* led to significant growth defects and generation of anucleate cells (Appendix Fig S10). Appearance of anucleate cells is an important sign of a defect in chromosome biology, thus suggesting that these three genes are involved in chromosome replication or segregation. SPD_0827 shows 33% identity with initiation control protein YabA of *Bacillus subtilis*, which interacts with DnaN and DnaA, and acts as a negative regulator of replication initiation (Noirot-Gros *et al*, 2002; Goranov *et al*, 2009). We thus named SPD_0827 to YabA. To test the function of *yabA* in *S. pneumoniae*, a deletion mutant was made by erythromycin marker replacement. The *yabA* deletion (Δ*yabA*) showed a significantly reduced growth rate compared to the wild type (Appendix Fig S12A) and displayed longer chains with frequent anucleate cells (Appendix Fig S12C). To test whether *S. pneumoniae* YabA is also a negative regulator of initiation of DNA replication, we determined the *oriC-ter* ratio using real-time quantitative PCR (qPCR). As shown in Appendix Fig S12D, the *oriC-ter* ratio was significantly higher in Δ*yabA* indicative of over-initiation, strongly suggesting a similar function as *B. subtilis* YabA.

When making a list of known genes involved in pneumococcal chromosome biology (Table 1), we noticed that *dnaB* and *dnaD*, two known bacterial DNA replication proteins (Smits *et al*, 2011; Briggs *et al*, 2012), are not annotated in *S. pneumoniae* D39. BlastP analyses showed that *spd_1405* and *spd_1522* might be coding for DnaD and DnaB, respectively. SPD_1405 showed 30% identity with DnaD of *B. subtilis,* and thus, we named *spd_1405* to *dnaD*. SPD_1522 has 389 amino acids (aa), and the N-terminal 1–149 aa-long domain showed 19.8% identity with domain I of DnaB of *B. subtilis*, whereas aa 206–379 showed 45.7% identity with domain II. DnaB of *B. subtilis* (472 aa) is longer than SPD_1522 of *S. pneumoniae* D39 (389 aa), because the former contains a degenerated middle DDBH2 domain (Briggs *et al*, 2012). Additionally, the arrangement of the neighboring genes of *S. pneumoniae dnaB (spd_1522)*, *dnaI*, and *nrdR* is the same in *B. subtilis.* Based on these observations, we named *spd_1522* to *dnaB*.

It was reported that DnaD and DnaB are recruited to the chromosome by DnaA and play important roles in chromosome replication initiation in *B. subtilis* (Smits *et al*, 2011). To test the function of *S. pneumoniae* DnaD and DnaB, we constructed $Zn^{2+}$-inducible depletion strains ($P_{Zn}$-*dnaD*; $P_{Zn}$-*dnaB*), because efforts to make deletion mutants failed. In the absence of 0.1 mM $Zn^{2+}$, the depletion strains showed significant growth defects (Appendix Fig S12A), confirming their essentiality. If DnaB and DnaD indeed play a role in replication initiation, repression of them should lead to a decrease in the *oriC-ter* ratio. Indeed, the *oriC-ter* ratio of cells in absence of $Zn^{2+}$ was significantly lower than in the presence of $Zn^{2+}$ (Appendix Fig S12D). Together, we identified and annotated *yabA, dnaD, and dnaB* and confirmed their function in pneumococcal DNA replication.

## SPD_1416 and SPD_1417 are involved in peptidoglycan precursor synthesis

We found that CRISPRi strains with sgRNA targeting hypothetical genes *spd_1416* or *spd_1417* showed significant growth retardation and morphological abnormality, such as heterogeneous cell size and elongated and enlarged cells with multiple incomplete septa (Appendix Fig S10). These manifestations mirrored what we observed upon inhibiting the expression of genes known to be involved in peptidoglycan (PG) synthesis (Appendix Fig S8). Consistent with the essentiality of these two genes as suggested by Tn-seq, we were unable to obtain deletion mutants of *spd_1416* or *spd_1417* after multiple attempts. To confirm that these genes are essential for pneumococcal growth, we constructed merodiploid strains of *spd_1416* and *spd_1417* by inserting a second copy of each gene fused to *gfp* (encoding a monomeric superfolder GFP) at their N-terminus (referred as *gfp-spd_1416* or *gfp-spd_1417*) or C-terminus (referred as *spd_1416-gfp* or *spd_1417-gfp*). These *gfp* fusions were integrated at the ectopic *bgaA* locus under the control of the zinc-inducible promoter, $P_{Zn}$. In the presence of $Zn^{2+}$, we could delete the native *spd_1416* or *spd_1417* gene by allelic replacement in the $P_{Zn}$-*gfp-spd_1417* or $P_{Zn}$-*gfp-spd_1416* genetic background. When transforming in the $P_{Zn}$-*spd_1417-gfp* genetic background, we did not obtain erythromycin resistant colonies, indicating that the C-terminal GFP fusion of SPD_1417 is not functional. Note that we could not replace *spd_1416* or *spd_1417* in the wild type in the presence of $Zn^{2+}$. While both the *spd_1416* and

*spd_1417* mutants behaved normally in the presence of $Zn^{2+}$, severe growth retardation was observed in the absence of $Zn^{2+}$ (Fig 4A). Together, these lines of evidence demonstrate that both *spd_1416* and *spd_1417* are essential genes.

Morphological analysis by light microscopy of bacterial cells upon depletion of *gfp-spd_1416* or *gfp-spd_1417* confirmed the morphological changes as observed in the CRISPRi knockdowns (Fig 4B). The *gfp-spd_1416* or *gfp-spd_1417* cells were further analyzed using freeze-substitution electron microscopy (Fig 4C). This showed the presence of elongated cells and the frequent formation of multiple septa per cell, in contrast to wild-type D39 cells which showed the typical diplococcal shape. Note that the mild sample preparation used in our freeze-substitution EM protocol also preserved the capsule, which can be readily lost during traditional EM sample preparation (Hammerschmidt *et al*, 2005). BlastP analysis shows that SPD_1416 contains a Mur-ligase domain with 36% sequence identity with MurT of *Staphylococcus aureus,* whereas SPD_1417 possesses a glutamine amidotransferase domain with 40% sequence identity with GatD of *S. aureus.* MurT and GatD, two proteins involved in staphylococcal cell wall synthesis (Figueiredo *et al*, 2012; Munch *et al*, 2012), form a complex to perform the amidation of the D-glutamic acid in the stem peptide of PG. It was previously reported that recombinant MurT/GatD of *S. pneumoniae* R6, purified from *E. coli*, indeed can amidate glutamate lipid II into iso-glutamine lipid II *in vitro* (Zapun *et al*, 2013). Therefore, we named *spd_1416* to *murT* and *spd_1417* to *gatD*. It is interesting to note that while MurT or GatD depletion strains in *S. aureus* showed reduced growth, cells exhibited normal cell morphologies (Figueiredo *et al*, 2012), in contrast to the strong morphological defects observed in *S. pneumoniae* D39.

MurT and GatD contain no membrane domain or signal peptide, and are thus predicted to be cytoplasmic proteins. However, fluorescence microcopy of the N-terminal GFP fused to MurT or GatD showed that they are partially membrane localized (Fig 4D). In-gel fluorescence imaging showed that GFP-MurT and GFP-GatD were correctly expressed without any detectable proteolytic cleavage (Appendix Fig S13). Since *in vitro* assays demonstrated that glutamate lipid II, which is anchored to the membrane by the bactoprenol hydrocarbon chain of lipid II, is a substrate of the MurT/GatD amidotransferase complex, it is reasonable to assume that membrane localization of MurT or GatD is caused by recruitment to the membrane-bound substrate. Indeed, amidation of the glutamic acid at position 2 of the peptide chain most likely occurs after formation of lipid-linked PG precursors (Rajagopal & Walker, 2016).

## CRISPRi revealed novel pneumococcal genes involved in teichoic acid biosynthesis

CRISPRi-based repression of hypothetical essential genes *spd_1197* and *spd_1198* led to significant growth defects, and microscopy revealed chained cells with abnormal shape and size (Appendix Fig S10). Some of the cells were elongated and enlarged. These phenotypes are consistent with the typical morphological changes caused by repression of genes in teichoic acid (TA) biosynthesis (Appendix Fig S9). In accordance with this, analysis of the genetic context of *spd_1197* and *spd_1198* showed that they are in the *lic3* region, which was predicted to be a pneumococcal TA gene cluster

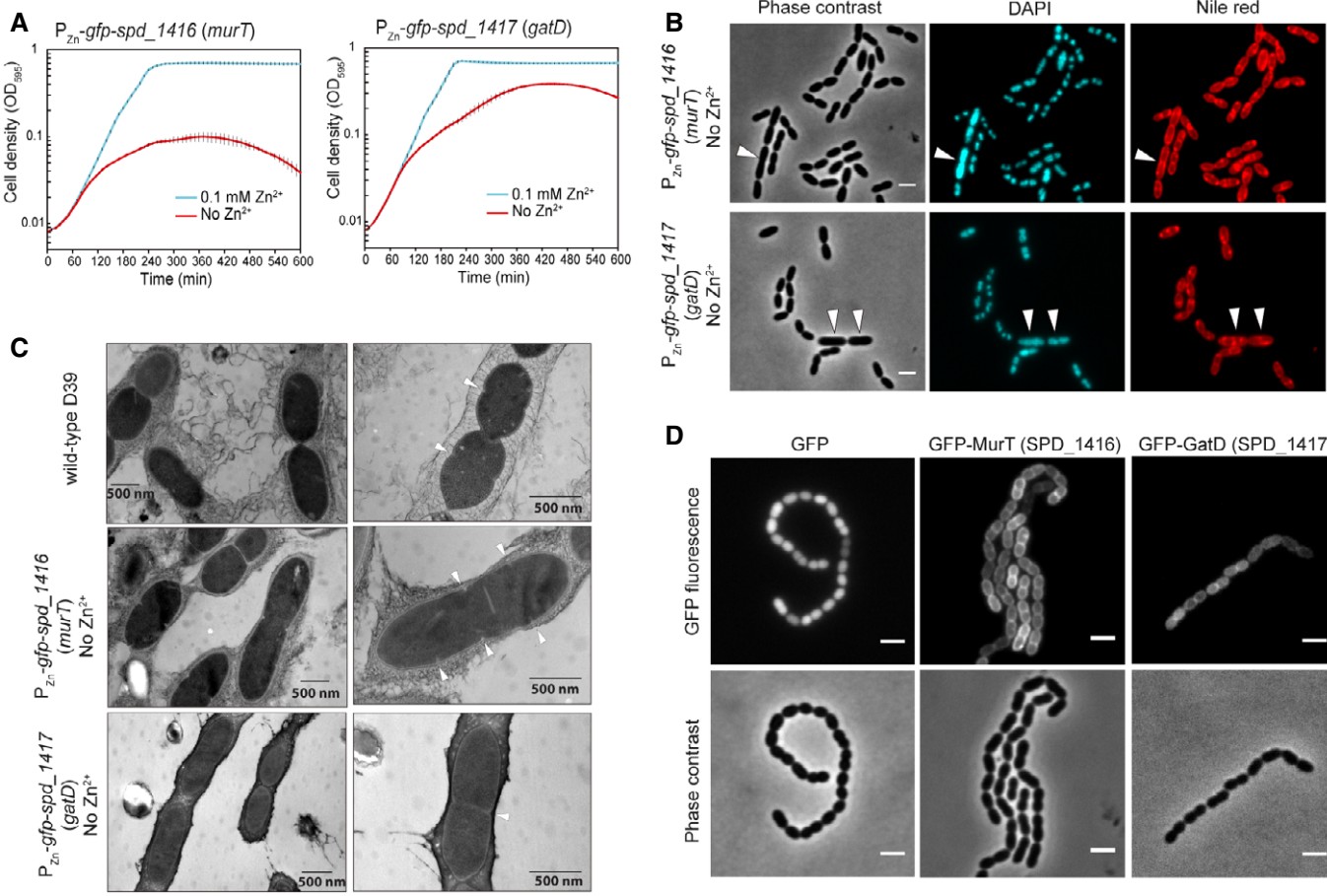

**Figure 4.  Identification of peptidoglycan synthesis genes *spd_1416* (*murT*) and *spd_1417* (*gatD*).**

A  Growth curves of depletion strains P$_{Zn}$-*gfp-spd_1416* (*murT*) and P$_{Zn}$-*gfp-spd_1417* (*gatD*), in C+Y medium with (cyan) or without (red) 0.1 mM Zn$^{2+}$. The values represent averages of three replicates with SEM.

B  Microscopy of cells from panel (A) after incubating in C+Y medium without Zn$^{2+}$ for 2.5 h. Representative micrographs of phase contrast, DAPI, and Nile red are shown. Scale bars = 2 μm. White arrows point to elongated and enlarged cells.

C  Electron micrographs of the same samples as in panel (B) and wild-type *S. pneumoniae* D39 as reference. Note that depletion of *spd_1416* or *spd_1417* resulted in elongated cells. Septa are pointed with white arrows.

D  Localization of GFP-MurT and GFP-GatD. Micrographs of GFP signal (upper panel) and phase contrast (lower panel) are shown. Scale bars = 2 μm. *Streptococcus pneumoniae* D39 with free GFP showing cytoplasmic localization was included as reference.

(Kharat *et al*, 2008; Denapaite *et al*, 2012). Similar to the approach described above, we generated Zn$^{2+}$-inducible C-terminal GFP fusions to SPD_1197 and SPD_1198, integrated these ectopically at the *bgaA* locus, and then deleted the native *spd_1197* or *spd_1198* genes in the presence of Zn$^{2+}$. Plate reader assays showed strong growth impairment in the absence of Zn$^{2+}$ (Fig 5A), suggesting their essentiality. In line with this, we were unable to replace these genes with an erythromycin resistance marker in the wild-type background in either the absence or presence of Zn$^{2+}$. Consistent with the phenotypes of the CRISPRi screen, the zinc-depletion strains showed similar morphological defects with cells in chains and elongated or enlarged cell shape and size (Fig 5B). EM analysis of depleted cells also revealed uneven distribution of multiple septa within a single cell, increased extracellular material and a rough cell surface (Fig 5C).

SPD_1198 contains 11 predicted transmembrane (TM) helices, while SPD_1197 has 2 predicted TM segments with a C-terminal extracytoplasmic tail. In-gel fluorescence showed that SPD_1197-GFP was mainly produced as a full-length product. The SPD_1198-GFP fusion, however, showed clear signs of protein degradation (Appendix Fig S13). Nevertheless, we performed fluorescence microscopy to determine their localizations. In agreement with the prediction, SPD_1197-GFP and SPD_1198-GFP are clearly localized to the membrane (Fig 5D).

Phosphorylcholine is an essential component of pneumococcal TA, and for this reason, a phosphorylcholine antibody is frequently used to detect *S. pneumoniae* TA (Vollmer & Tomasz, 2001; Wu *et al*, 2014). To explore whether SPD_1197 and SPD_1198 indeed play a role in TA synthesis, we performed Western blotting to detect phosphorylcholine-decorated TA using whole-cell lysates (Fig 5E). Cells of strains P$_{Zn}$-*spd_1197-gfp* and P$_{Zn}$-*spd_1198-gfp* were grown in the presence or absence of 0.1 mM Zn$^{2+}$. As controls, we depleted expression of three genes involved in PG synthesis (*murT*, *gatD*, and *pbp2x*). As shown in Fig 5E, Zn$^{2+}$ did

**Figure 5.   Newly identified genes of the teichoic acid biosynthesis pathway: *spd_1198* (*tarP*) and *spd_1197* (*tarQ*) are involved in precursor polymerization.**

A   Growth curves of depletion strains $P_{Zn}$-*spd_1197-gfp* (*tarQ*) and $P_{Zn}$-*spd_1198-gfp* (*tarP*) in C+Y medium with (cyan) or without (red) 0.1 mM $Zn^{2+}$. The values represent averages of three replicates with SEM.

B   Microscopy of strains as in panel (A) after incubation in C+Y medium without $Zn^{2+}$ for 2.5 h. Representative micrographs are shown. Scale bars = 2 μm. White arrows point to elongated and enlarged cells. Note that depletion of *spd_1197* or *spd_1198* led to long-chain formation of cells.

C   Electron micrographs of the same samples as in panel (B) with wild-type *S. pneumoniae* D39 as reference. Arrowheads point to the septa of cells.

D   Localization of TarQ-GFP, TarP-GFP, with C-terminal fused monomeric GFP. GFP signal (upper panel) and phase contrast (lower panel) are shown. Scale bars = 2 μm.

E   Western blotting to detect phosphorylcholine-containing molecules of *S. pneumoniae*. Whole-cell lysates were separated with SDS–PAGE, and phosphorylcholine-containing molecules were detected by phosphorylcholine antibody TEPC-15. Smaller bands caused by depletion of *tarQ* (*spd_1197*) or *tarP* (*spd_1198*) are indicated by asterisks. Note that for *tarQ*, *tarP*, *murT*, and *gatD*, $Zn^{2+}$-inducible strains were used, and for *pbp2X*, a CRISPRi strain was used.

F   Model for TarP/TarQ function in precursor polymerization of the teichoic acid biosynthesis pathway in *S. pneumoniae*. Steps of biosynthesis of repeat units (RU), decoration of RU with choline, and polymerization of the precursor are shown.

not influence TA synthesis of the *S. pneumoniae* D39 wild-type (WT) strain, and the four main TA bands are clearly visible, migrating in the range between 15 and 25 kDa consistent with

previous reports (Wu *et al*, 2014). In contrast, cells depleted for SPD_1197 or SPD_1198 displayed a different pattern and the 4 main bands around 15 and 25 kDa were missing or much weaker,

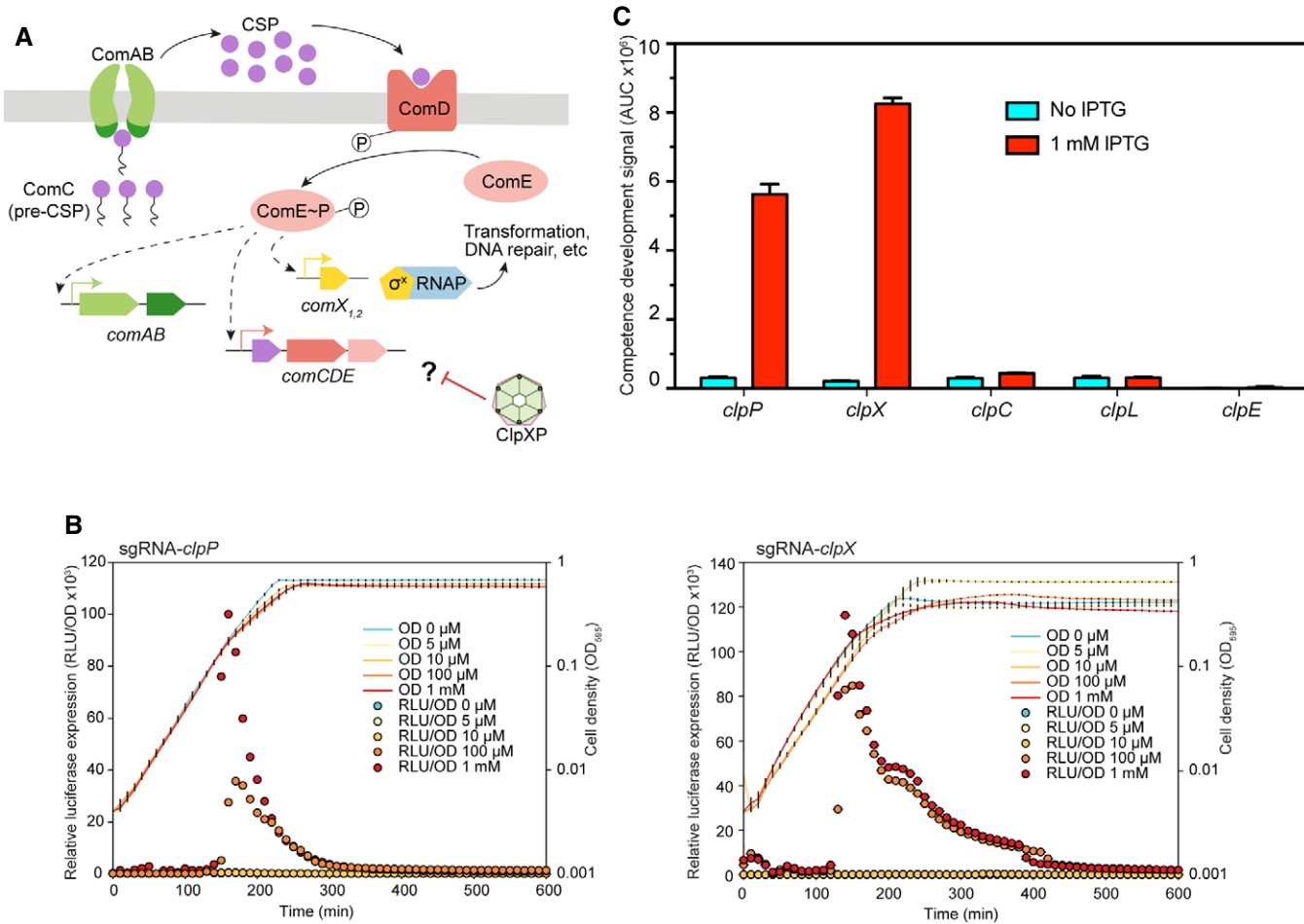

**Figure 6.  The ATPase ClpX and the ClpP protease repress competence development.**

A    Regulatory network of the competence pathway. Competence is induced when the *comC*-encoded competence-stimulating peptide (CSP) is recognized, cleaved, and exported by the membrane transporter (ComAB). Accumulation of CSP then stimulates its receptor (membrane-bound histidine-kinase ComD), which subsequently activates ComE by phosphorylation, which in turn activates the expression of the so-called early competence genes. One of them, *comX*, codes for a sigma factor, which is responsible for the activation of over 100 competence genes, including those required for transformation and DNA repair. Here, we show that the ATPase subunit ClpX works together with the protease ClpP, repressing competence, probably by negatively controlling the basal protein level of the competence regulatory proteins, but the exact mechanism is unknown (question mark).

B    Repression of *clpP* or *clpX* by CRISPRi triggers competence development. Activation of competence system is reported by the *ssbB_luc* transcriptional fusion. Detection of competence development was performed in C+Y medium at a pH in which natural competence of the wild-type strain is uninduced. IPTG was added to the medium at the beginning at different final concentrations (0, 5 μM, 10 μM, 100 μM, 1 mM). Cell density (OD$_{595}$) and luciferase activity of the bacterial cultures were measured every 10 min. The values represent averages of three replicates with SEM.

C    Influence of repression of *clpP*, *clpX*, *clpC*, *clpL*, and *clpE* on competence development. AUC (area under the curve) of the relative luciferase expression curve in panel (B) (1 mM IPTG and no IPTG) and Fig EV5 was calculated and used to represent the competence development signal. The values represent averages of three replicates with SEM.

while multiple bands with a size smaller than 15 kDa appeared. TA of *S. pneumoniae*, including wall teichoic acid (WTA) and membrane-anchored lipoteichoic acid (LTA), are polymers with identical repeating units (RU) (Fischer *et al*, 1993). Addition of one RU can lead to about a 1.3 kDa increase in molecular weight (Gisch *et al*, 2013). Interestingly, the weight interval between the extra smaller bands from bacterial cells with depleted SPD_1197 or SPD_1198 seemed to match the molecular weight of the RU, suggesting that SPD_1197 and SPD_1198 play a role in TA precursor polymerization. Although repression of the genes associated with peptidoglycan synthesis (*murT*, *gatD* and *pbp2x*) made the 4 main TA bands weaker, the pattern of the TA bands was not changed. Likely, the reduction in the TA of these three strains is

due to the reduction in peptidoglycan, which constitutes the anchor for wall TA. Additionally, a CRISPRi strain targeting *tarI* of the *lic1* locus, which is involved in an early step of TA precursor synthesis, was included as a control. Note that *tarI* is cotranscribed with the other four genes of the *lic1* locus, including *tarJ*, *licA*, *licB*, and *licC*. Likely, CRISPRi knockdown of *tarI* will repress transcription of the entire *lic1* locus and thus block the synthesis of TA precursors. In line with this, we observed a reduction in the total amount of teichoic acid chains when *tarI* was repressed by CRISPRi (Appendix Fig S14).

The TA chains of *S. pneumoniae* are thought to be polymerized before they are transported to the outside of the membrane by the flippase TacF (Damjanovic *et al*, 2007), and so far it is not known

which protein(s) function(s) as TA polymerase (Denapaite *et al*, 2012). In line with SPD_1198 being the TA polymerase, homology analysis shows that it contains a predicted polymerase domain. The large cytoplasmic part of SPD_1197 may aid in the assembly of the TA biosynthetic machinery by protein–protein interactions (Denapaite *et al*, 2012). Together, we here show that SPD_1197 and SPD_1198 are essential for growth and we suggest that they are responsible for polymerization of TA chains (Fig 5F). Consistent with the nomenclature used for genes involved in TA biosynthesis, we named *spd_1198 tarP* (for teichoic acid ribitol polymerase) and *spd_1197 tarQ* (in operon with *tarP*, sequential alphabetical order). Whether TarP and TarQ interact and function as a complex remains to be determined.

### The essential ATPase ClpX and the protease ClpP repress competence development

We wondered whether we could also employ CRISPRi to probe gene regulatory networks in which essential genes play a role. An important pathway in *S. pneumoniae* is development of competence for genetic transformation, which is under the control of a well-studied two-component quorum sensing signaling network (Claverys *et al*, 2009). Several lines of evidence have shown that the highly conserved ATP-dependent Clp protease, ClpP, in association with an ATPase subunit (either ClpC, ClpE, ClpL, or ClpX), is involved in regulation of pneumococcal competence (Charpentier *et al*, 2000; Chastanet *et al*, 2001) (Fig 6A). Identification of the ATPase subunit responsible for ClpP-dependent repression of competence was hampered because of the essentiality, depending on the growth medium and laboratory conditions, of several *clp* mutants including *clpP* and *clpX* (Chastanet *et al*, 2001). To address this issue, we employed CRISPRi and constructed sgRNAs targeting *clpP, clpC, clpE, clpL,* and *clpX*. Competence development was quantified using a *luc* construct, driven by a competence-specific promoter (Slager *et al*, 2014). As shown in Fig 6B, when expression of ClpP or ClpX was repressed by addition of IPTG, competence development was enhanced, while depleting any of the other ATPase subunits (ClpC, ClpE, and ClpL) had no effect on competence (Figs 6C and EV5). This shows that ClpX is the main ATPase subunit responsible for ClpP-dependent repression of competence.

## Discussion

Here, we developed an IPTG-inducible CRISPRi system to study essential genes in *S. pneumoniae* (Fig 1). In addition, we adopted a simple and efficient one-step sgRNA engineering strategy using infusion cloning. This approach resulted in ~89% positive sgRNA clones after a single round of transformation, thus enabling high-throughput cloning of sgRNAs.

Growth analysis of the CRISPRi strains targeting the 348 potentially essential genes showed that individual repression of 73% of the targeted genes led to growth phenotypes, using a stringent cutoff for phenotype detection (Figs 2B and C, and EV2). There could be several reasons why CRISPRi knockdown of the remaining 94 genes did not cause a detectable growth phenotype. Tn-seq sometimes incorrectly assigns an essential function to non-essential genes (van

Opijnen *et al*, 2009; van Opijnen & Camilli, 2013). Also, Tn-seq relies on a round of growth on blood agar plates, while our CRISPRi phenotypes were only assayed in liquid C+Y medium. Additionally, we used stringent cutoffs for phenotype definition, which will miss genes with mild growth or lysis phenotypes. Certain genes might also not be repressed well enough by CRISPRi to show a phenotype (in case of stable proteins that only require a few molecules for growth). This can be for instance caused when the sgRNA targets a PAM site far away from the transcription start site, when there is poor access of the sgRNA-dCas9 complex to the target DNA or when there are polar effects within the operon alleviating the essentiality. We can also not exclude a suppressor mutation arising in some of the "No phenotype" CRISPRi strains, as most CRISPRi knockdowns with growth phenotypes eventually grew out to the same final OD and contain a loss-of-function mutation in the coding sequence of *dcas9* (Fig EV4).

Based on analysis of the CRISPRi knockdowns, several previously "hypothetical" genes could be functionally characterized and annotated. For instance, combined with BlastP analysis and determination of *oriC-ter* ratios, we could annotate the pneumococcal primosomal machinery, including DnaA, DnaB, DnaC, DnaD, DnaG, and DnaI (Table 1, Appendix Figs S2 and S12). Note that *spd_2030 (dnaC)* was mis-annotated as *dnaB* in several databases, such as in NCBI (ProteinID: ABJ54728), KEGG (Entry: SPD_2030), Uniprot (Entry: A0A0H2ZNF7), which may be due to the different naming of primosomal proteins in *E. coli* and *Bacillus subtilis* (Smits *et al*, 2011; Briggs *et al*, 2012). By characterizing CRISPRi-based knockdowns with cell morphology defects, we identified four essential cell wall biosynthesis genes (*murT, gatD, tarP,* and *tarQ*), which are promising candidates for future development of novel antimicrobials.

This work and other studies highlight that high-throughput phenotyping by CRISPRi is a powerful approach for hypothesis-forming and functional characterization of essential genes (Peters *et al*, 2016). We also show that CRISPRi can be used to unravel gene regulatory networks in which essential genes play a part (Fig 6). While we shed light on the function of just several previously uncharacterized essential genes, the here-described library contains richer information that needs to be further explored. In addition, CRISPRi screens can be used for mechanism of action (MOA) studies with new bioactive compounds. Indeed, CRISPRi was recently successfully employed to show that *B. subtilis* UppS is the molecular target of compound MAC-0170636 (Peters *et al*, 2016). We anticipate that the here-described pneumococcal CRISPRi library can function as a novel drug target discovery platform, can be applied to explore host–microbe interactions, and will provide a useful tool to increase our knowledge concerning pneumococcal cell biology.

## Materials and Methods

### Strains, growth conditions, and transformation

Oligonucleotides are shown in Dataset EV4 and strains in Appendix Table S1. *Streptococcus pneumoniae* D39 and its derivatives were cultivated in C+Y medium, pH = 6.8 (Slager *et al*, 2014) or Columbia agar with 2.5% sheep blood at 37°C. Transformation of *S. pneumoniae* was performed as previously described (Martin *et al*, 2000), and CSP-1 was used to induce competence.

Transformants were selected on Columbia agar supplemented with 2.5% sheep blood at appropriate concentrations of antibiotics (100 μg/ml spectinomycin, 250 μg/ml kanamycin, 1 μg/ml tetracycline, 40 μg/ml gentamycin, 0.05 μg/ml erythromycin). For construction of depletion strains with the $Zn^{2+}$-inducible promoter, 0.1 mM $ZnCl_2$/0.01 mM $MnCl_2$ was added to induce the ectopic copy of the target gene (mentioned as 0.1 mM $Zn^{2+}$ for convenience). Working stock of the cells, called "T2 cells", were prepared by growing the cells in C+Y medium to $OD_{600}$ 0.4, and then resuspending the cells with equal volume of fresh medium with 17% glycerol.

*Escherichia coli* MC1061 was used for subcloning of plasmids, and competent cells were prepared by $CaCl_2$ treatment. The *E. coli* transformants were selected on LB agar with appropriate concentrations of antibiotics (100 μg/ml spectinomycin, 100 μg/ml ampicillin, 50 μg/ml kanamycin).

## Construction of an IPTG-inducible CRISPRi system in *S. pneumoniae*

*Streptococcus pyogenes dcas9* (*dcas9sp*) was obtained from Addgene (Addgene #44249, Qi *et al*, 2013) and subcloned into plasmid pJWV102 (Veening laboratory collection) with the IPTG-inducible promoter $P_{lac}$ (Sorg, 2016) replacing $P_{Zn}$, resulting in plasmid pJWV102-$P_{lac}$-*dcas9sp*. pJWV102-$P_{lac}$-*dcas9sp* was integrated into the *bgaA* locus in *S. pneumoniae* D39 by transformation. To control $P_{lac}$ expression, a codon-optimized *E. coli lacI* gene driven by the constitutive promoter PF6 was inserted at the *prsA* locus in *S. pneumoniae* D39 (Sorg, 2016), leading to the construction of strain DCI23. DCI23 was used as the host strain for the insertion of gene-specific sgRNAs and enables the CRISPRi system. The DNA fragment encoding the single-guide RNA targeting luciferase (sgRNA*luc*) was ordered as a synthetic DNA gBlock (Integrated DNA Technologies) containing the constitutive P3 promoter (Sorg *et al*, 2015). The sgRNA*luc* sequence is transcribed directly after the +1 of the promoter and contains 19 nucleotides in the base-pairing region, which binds to the non-template (NT) strand of the coding sequence of luciferase, followed by an optimized single-guide RNA (Chen *et al*, 2013) (Fig EV1A). Then, the sgRNA*luc* with P3 promoter was cloned into pPEP1 (Sorg *et al*, 2015) with removing the chloramphenicol resistance marker (pPEPX) leading to the production of plasmid pPEPX-P3-sgRNA*luc*, which integrates into the region between *amiF* and *treR* of *S. pneumoniae* D39. The pPEPX-P3-sgRNA*luc* is used as the template for generation of other sgRNAs by infusion cloning or by the inverse PCR method. The *lacI* gene with gentamycin resistance marker and flanked *prsA* regions was subcloned into pPEPY (Veening laboratory collection), resulting in plasmid pPEPY-PF6-*lacI*. This plasmid can be used to amplify *lacI* and integrate it at the *prsA* locus while selecting for gentamycin resistance. The entire pneumococcal CRISPRi system, consisting of plasmids pJWV102-$P_{lac}$-*dcas9sp*, pPEPY-PF6-*lacI*, and pPEPX-P3-sgRNA*luc*, is available from Addgene (ID 85588, 85589, and 85590, respectively).

## Selection of essential genes

To identify each gene's contribution to fitness for basal level growth, we performed Tn-seq in *S. pneumoniae* D39 essentially as described before (Zomer *et al*, 2012; Burghout *et al*, 2013), but with growing cells in C+Y medium at 37°C. Possibly essential genes were identified using ESSENTIALS (Zomer *et al*, 2012). Based on that, we included all the identified essential genes and added extra essential genes identified in serotype 4 strain TIGR4 (van Opijnen *et al*, 2009; van Opijnen & Camilli, 2012). Note that in the Tn-seq study of 2012, fitness of each gene under 17 *in vitro* and 2 *in vivo* conditions was determined and genes were grouped into different classes (van Opijnen & Camilli, 2012). Finally, 391 genes were selected (Dataset EV1).

## Oligonucleotides for the CRISPRi library

The 20-nt guide sequences of the sgRNAs targeting different genes were selected with CRISPR Primer Designer (Yan *et al*, 2015). Briefly, we searched within the coding sequence of each essential gene for a 14-nt specificity region consisting of the 12-nt "seed" region of the sgRNA and GG of the 3-nt PAM (GGN). sgRNAs with more than one binding site within the pneumococcal genome, as determined by a BLAST search, were discarded. Next, we took a total length of 21 nt (including the +1 of the P3 promoter and 20 nt of perfect match to the target) and the full-length sgRNA's secondary structure was predicted using ViennaRNA (Lorenz *et al*, 2011), and the sgRNA sequence was accepted if the dCas9 handle structure was folded correctly (Larson *et al*, 2013). We chose the guide sequences as close as possible to the 5′ end of the coding sequence of the targeted gene (Qi *et al*, 2013). The sequences of the sgRNAs (20 nt) are listed in Dataset EV3.

## Cloning of sgRNA

We used infusion cloning instead of inverse PCR recommended by Larson *et al* (2013) because significantly higher cloning efficiencies were obtained with infusion cloning. Two primers, sgRNA_inF_plasmid_linearize_R and sgRNA_inF_plasmid_linearize_F, were designed for linearization of plasmid pPEPX-P3-sgRNA*luc*. These two primers bind directly upstream and downstream of the 19-bp guide sequence for *luc*. To fuse the 20-nt new guide sequence into the linearized vector, two 50-nt complementary primers were designed for each target gene. Each primer contains 15 nt at one end, overlapping with the sequence on the 5′ end of the linearized vector, followed by the 20-nt specific guide sequence for each target gene; and 15 nt overlapping with the sequence on the 3′ end of the linearized vector (Fig EV2A). The two 50-nt complementary primers were annealed in TEN buffer (10 mM Tris, 1 mM EDTA, 100 mM NaCl, pH 8) by heating at 95°C for 5 min and cooling down to room temperature. The annealed product was fused with the linearized vector using the Quick-Fusion Cloning kit (BiMake, Cat. B22612) according to the manufacturer with the exception of using only one half of the recommended volume per reaction. Each reaction was directly used to transform competent *S. pneumoniae* D39 strain DCI23.

## Luciferase assay

*Streptococcus pneumoniae* strains XL28 and XL29 were grown to $OD_{600} = 0.4$ in 5-ml tubes at 37°C and then diluted 1:100 in fresh C+Y medium with or without 1 mM IPTG. Then, in triplicates, 250-μl diluted bacterial culture was mixed with 50 μl of 6× luciferin solution in C+Y medium (2.7 mg/ml, D-Luciferin sodium salt,

SYNCHEM OHG) in 96-well plates (Polystyrol, white, flat, and clear bottom; Corning). Optical density at 595 nm ($OD_{595}$) and luminescence signal were measured every 10 min for 10 h using a Tecan Infinite F200 Pro microtiter plate reader.

## Growth assays

For growth curves of strains of the CRISPRi library, T2 cells were thawed and diluted 1:1,000 into fresh C+Y medium with or without 1 mM IPTG. Then, 300 µl of bacterial culture was added into each well of 96-well plates. $OD_{595}$ was measured every 10 min for 18 h with a Tecan Infinite F200 Pro microtiter plate reader. Specially, for the data shown in Fig 3, Appendix Figs S10 and S11, T2 cells were diluted 1:100 in C+Y medium. For growth assays of the depletion strains with the $Zn^{2+}$-inducible promoter, T2 cells were thawed and diluted 1:100 into fresh C+Y medium with or without 0.1 mM $Zn^{2+}$.

## Detection of teichoic acids

### Sample preparation

T2 cells of *S. pneumoniae* strains were inoculated into fresh C+Y medium with 0.1 mM $Zn^{2+}$ by 1:50 dilution, and then grown to $OD_{600}$ 0.15 at 37°C. Cells were collected at 8,000 *g* for 3 min and resuspended with an equal volume of fresh C+Y medium without $Zn^{2+}$. Bacterial cultures were diluted 1:10 into C+Y with or without 0.1 mM $Zn^{2+}$ or 1 mM IPTG (for CRISPRi strains) and then incubated at 37°C. When $OD_{600}$ reached 0.3, cells were centrifuged at 8,000 *g* for 3 min. The pellets were washed once with cold TE buffer (10 mM Tris–Cl, pH 7.5; 1 mM EDTA, pH 8.0), and resuspended with 150 µl of TE buffer. Cells were lysed by sonication.

### Detection of teichoic acid with phosphoryl choline antibody MAb TEPC-15

Protein concentration of the whole-cell lysate was determined with the DC protein assay kit (Bio-Rad Cat. 500-0111). Whole-cell lysates were mixed with equal volumes of 2× SDS protein loading buffer (100 mM Tris–HCl, pH 6.8; 4% SDS; 0.2% bromophenol blue; 20% glycerol; 10 mM DTT) and boiled at 95°C for 5 min. 2 µg of protein was loaded, followed by SDS–PAGE on a 12% polyacrylamide gel with cathode buffer (0.1 M Tris, 0.1 M tricine, 0.1% SDS) on top of the wells and anode buffer (0.2 M Tris/Cl, pH 9.9) in the bottom. After electrophoresis, samples in the gel were transferred onto a polyvinylidene difluoride (PVDF) membrane as described (Minnen *et al*, 2011). Teichoic acid was detected with anti-PC specific monoclonal antibody TEPC-15 (M1421, Sigma) by 1:1,000 dilution as first antibody, and then with anti-mouse IgG HRP antibody (GE Healthcare UK Limited) with 1:5,000 dilution as second antibody. The blots were developed with ECL prime Western blotting detection reagent (GE Healthcare UK Limited), and the images were obtained with a Bio-Rad imaging system.

## Microscopy

To detect the morphological changes after knockdown of the target genes, strains in the CRISPRi library were induced with IPTG and depletion strains were incubated in C+Y medium without $Zn^{2+}$, stained with DAPI (DNA dye) and Nile red (membrane dye), and then studied by fluorescence microscopy. Specifically, 10 µl of thawed T2 cells was added into 1 ml of fresh C+Y medium, with or without 1 mM IPTG, in a 1.5-ml Eppendorf tube, followed by 2.5 h of incubation at 37°C. After that, 1 µl of 1 mg/ml Nile red was added into the tube and cells were stained for 4 min at room temperature. Then, 1 µl of 1 mg/ml DAPI was added and the mix was incubated for one more minute. Cells were spun down at 8,000 *g* for 2 min, and then, the pellets were suspended with 30 µl of fresh C+Y medium. 0.5 µl of cell suspension was spotted onto a PBS agarose pad on microscope slides. DAPI, Nile red, and phase contrast images were acquired with a Deltavision Elite (GE Healthcare, USA). Microscopy images were analyzed with ImageJ.

For fluorescence microscopy of strains containing zinc-inducible GFP fusions, strains were grown in C+Y medium to $OD_{600}$ 0.1, followed by 10 times dilution in fresh C+Y medium with 0.1 mM $Zn^{2+}$. After 1 h of incubation, cells were spun down, washed with PBS, and resuspended in 50 µl PBS. 0.5 µl of cell suspension was spotted onto a PBS agarose pad on microscope slides. Visualization of GFP was performed as described previously (Kjos *et al*, 2015).

For electron microscopy, T2 cells of *S. pneumoniae* strains were inoculated into C+Y medium with 0.1 mM $Zn^{2+}$ and incubated at 37°C. When $OD_{600}$ reached 0.15, the bacterial culture was centrifuged at 8,000 *g* for 2 min. The pellets were resuspended into C+Y without $Zn^{2+}$ such that $OD_{600}$ was 0.015, and then, cells were incubated again at 37°C. Bacterial cultures were put on ice to stop growth when $OD_{600}$ reached 0.35. Cells were collected by centrifugation and washed once with distilled water. A small pellet of cells was cryo-fixed in liquid ethane using the sandwich plunge freezing method (Baba, 2008) and freeze-substituted in 1% osmium tetroxide, 0.5% uranyl acetate, and 5% distilled water in acetone using the fast low-temperature dehydration and fixation method (McDonald & Webb, 2011). Cells were infiltrated overnight with Epon 812 (Serva, 21045) and polymerized at 60°C for 48 h. 90-nm-thick sections were cut with a Reichert ultramicrotome and imaged with a Philips CM12 transmission electron microscope running at 90 kV.

## Competence assays

The previously described *ssbB_luc* competence reporter system, amplified from strain MK134 (Slager *et al*, 2014), was transformed into the CRISPRi strains (sgRNA*clpP*, sgRNA*clpX*, sgRNA*clpL*, sgRNA*clpE*, sgRNA*clpC*). Luminescence assays for detection of activation of competence system were performed as previously described (Slager *et al*, 2014). IPTG was added into C+Y medium (at a non-permissive pH for competence development) at the beginning of cultivation to different final concentrations.

## Data availability

Raw Tn-seq data are available at the SRA with accession number SRR5298192; the RNA-Seq data are available at the GEO database with accession number GSE89763.

**Expanded View** for this article is available online.

## Acknowledgements

We thank A. Zomer, P. Burghout, and P. Hermans for help with acquiring and analyzing the Tn-seq data. We thank K. Kuipers and M. I. de Jonge for

providing the TEPC-15 antibodies and for the TA Western blotting protocol. V. Benes and B. Haase (GeneCore, EMBL, Heidelberg) are thanked for sequencing support. XL is supported by China Scholarship Council (No. 201506210151). ADP was supported by a "Marie Curie IF" grant (call H2020-MSCA-IF-2014, number 657546). KK is supported by a NWO VENI fellowship (563.14.003). MK is supported by a grant from the Research Council of Norway (250976/F20). Work in the Veening laboratory is supported by the EMBO Young Investigator Program, a VIDI fellowship (864.12.001) from the Netherlands Organization for Scientific Research, Earth and Life Sciences (NWO-ALW), and ERC Starting Grant 337399-PneumoCell.

## Author contributions

XL and J-WV designed the study. XL, MK, AD, CG, SPK, JS, and JWV performed experiments and analyzed the data. KK performed electron microscopy. RAS developed the IPTG-inducible system. JS analyzed the RNA-Seq data and performed the CRISPRi growth analysis. XL, JS, MK, AD, CG, J-RZ, and J-WV wrote the manuscript with input from all authors. All authors read and approved the final manuscript.

## Conflict of interest

The authors declare that they have no conflict of interest.

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
