## [Review Process File · Molecular Systems Biology]

High-throughput CRISPRi phenotyping identifies new essential genes in *Streptococcus pneumoniae*

Xue Liu, Clement Gallay, Morten Kjos, Arnau Domenech, Jelle Slager, Sebastiaan P van Kessel, Kevin Knoops, Robin A Sorg, Jing-Ren Zhang, Jan-Willem Veening

Corresponding author: Jan-Willem Veening, University of Lausanne

Review timeline:

Submission date:	17 November 2016
Editorial Decision:	14 December 2016
Revision received:	03 March 2017
Editorial Decision:	03 April 2017
Revision received:	12 April 2017
Accepted:	13 April 2017

Editor: Maria Polychronidou

Transaction Report:

1st Editorial Decision

14 December 2016

Thank you again for submitting your work to Molecular Systems Biology. We have now heard back from the three referees who agreed to evaluate your study. As you will see below, the reviewers appreciate that the presented CRISPRi screen is a useful resource for the field. However, they raise several concerns, which we would ask you to address in a revision.

I think that the reviewers' recommendations are rather clear so there is no need to repeat the points listed below, but please let me know in case you would like to further discuss any specific point. One of the more fundamental issues raised by the reviewers, refers to the need to include additional controls.

REFeree REPORTS

Reviewer #1:

The study by Liu and coauthors is an elegant example of the adaptation of CRISPRi technology to the phenotypization of bacterial genes. This technology has previously been adapted to a few other bacteria, and the scale of the current work certainly merits significant attention. The authors focus their work on genes predicted to be essential by their transposon mutagenesis experiments, designing a library of sgRNAs that spans ~15% of the protein coding genes. From careful phenotypic analysis of several candidates the authors help elucidate genes involved in peptidoglycan and teichoic acid synthesis, and natural competence. The latter experiments are particularly striking, in part because of the quality of the negative controls provided by the non-participating subunits. My enthusiasm for the study is mollified by the general lack of key controls present in the study. For

instance, an adequate number of targets predicted to be dispensable in their transposon experiment could have served to assess the precision of their CRISPRi approach. Such controls could be easily included and would significantly improve the study, helping to assess the reliability of the CRISPRi method, and increasing the confidence of a few key observations. Provided with the adequate controls, I have no doubt the work will have a tremendous impact on the study of *Streptococcus pneumoniae*.

MAJOR COMMENTS:

Fig. 2. Given that polar effects have been well documented when using CRISPRi in bacteria, the authors should analyze their screening results in the context of the operon architecture. When targeting the same operon, what is the agreement of different sgRNAs? Such an analysis would be facilitated by including selected sgRNAs against operons that contain both essential and dispensable genes. Are there any cases where the CRISPRi effect on a known dispensable operon, extends into an essential operon? Quantifying and displaying the magnitude of such effects, for instance by performing RNAseq experiments on a couple of other genes, would help determine the degree of confidence that readers should have on the authors' data set. However, a simple analysis of the existing data is probably sufficient to satisfy this point, since there are a lot of genes that did not show a phenotype in their analysis.

Fig. 2C. Were any of the genes predicted to be dispensable further analyzed? Presumably some of these were particularly implicated in essential functions by the transposon experiments, yet appeared dispensable by CRISPRi. Understanding the source of the discrepancy, could help establish the reliability of the methods and the ultimate relevance of the screening results.

Fig. 3. Measuring growth curved is a central part of the study's phenotypic assessment, and would benefit from a set of genes pre-selected as controls. Is there a set of genes that are known to be dispensable and could be used as controls? This is particularly important given the documented polar effects of the CRISPRi system. Can sgRNAs against dispensable genes at the end of an operon be used to repress such genes without affecting the rest of the operon?

Fig. 4. Depletion of the GFP-tagged proteins should be demonstrated, to confirm the effects of the genetic manipulations. This could be ideally performed by immunoblotting against the GFP portion of the fusion proteins, compared to a control. Alternatively the strains could be compared by FACS, provided an adequate counterstain to gate on the bacteria.

Fig. 5E. I recognize that the monoclonal antibody against phosphorylcholine has previously been used; however, to my knowledge, the small species found in the TarP/Q knockdowns have not been documented. If I am wrong on this point, please reference such an observation appropriately. Alternatively, the authors should include a mutant of an upstream member of the pathway to fully identify the bands they claim to be repeating units, and thereby validate their assay.

Fig. 6B. As mentioned above, I believe the strength of these experiments is the lack of an effect in the case of the *clpC*, *clpE* and *clpL* strains. Finding a way to summarize the results from the growth curves in a bar graph could be used to bring these important controls into the main text.

MINOR COMMENTS:

lines 222-224: The sentence needs clarification. Was a different construct than the one described above used? It is not clear from the context what is being introduced by erythromycin selection.

line 266: Inability to replace the genes merely suggests essentiality; it does not "confirm" it.

Reviewer #2:

Summary

Liu et. al. constructed a genome scale library of essential gene knockdown strains by using CRISPR interference (CRISPRi) in *S. pneumoniae*, and identified phenotypes for the knockdown strains

using a systematic microscopy screen. The importance of this study is that it is the first CRISPRi screen in non-rod-shaped bacteria, as well as the first one in a pathogen, and will be a valuable resource for the research community. It further shows the utility of using CRISPRi as for studying essential gene function. The authors follow up on a number of phenotypes to ascribe function to essential genes without previous functional description. Several aspects of the manuscript should be clarified or improved, as described below.

General Comments

1. The overarching theme of this work is to ascribe function to poorly characterized genes that are essential to the organism. Therefore, it is incumbent on the authors to carefully describe both existing annotation and how the gene set to be analyzed was chosen.
 - a. which databases were used?1. The au
 - b. Line 347: please expand on the source and nature of the mis-annotation
 - c. In general, Tn-seq screens overestimate the number of essential genes. What criteria did you use for deciding on the set of genes to profile?
 - d. Lines 207-208: Please explain in the text how you determined which subset to phenotype.

2. Sequence similarity between investigated genes and other genes should be better described in the body of the text, so that the reader can distinguish between cases where sequence similarity might have sufficed to establish function and cases where the authors follow-up was essential. At present, sequence similarity is discussed only in the supplement. For example in the investigation of dnaB/dnaD and spd1405/spd1522 the sequence similarities and bioinformatics likely would have sufficed to establish function, whereas for tarP/tarQ, the CRISPRi phenotype and follow-up was essential. Example: In Line 202: "our data revealed" should be clarified, at least in part by bringing supplemental data about sequence similarity into the body of the text.

3. For the genes explored in detail, care should be taken to explicate the genomic context and enumerate the potential polar effects within operons. The recent work on the *B. subtilis* essential gene set showed that CRISPRi exhibits polarity both on downstream and upstream genes in an operon. Therefore, if the knockdown gene of interest is in an operon, the authors should mention the possibility that several genes in addition to the targeted gene may be responsible for the phenotype. One example of this issue is that the authors show the rpoA (alpha subunit of RNA polymerase) as an indicator of transcription. However, rpoA is in an operon with ribosomal proteins and the phenotype could be one resulting from those genes.

4. Various experiments are missing important controls. Examples:
 - a. Figure 1D: please add a control comparing WT (+luc, no-dCas9) to the dCas9 strain (+dCas9, +luc) to verify that Cas9 itself is not responsible for changes.
 - b. Lines 149-150: Is the sequence targeted by sgRNA_{luc} also found in the com operon (as a perfect or imperfect match)? This might explain the slight-but reproducible -knockdown of com genes in Fig. 1D. It would be a good idea to mention here that you ruled out perfect or near-perfect matches.
 - c. Lines 340-342: If you really are picking up suppressors during the growth of CRISPRi knockdown strains, I would expect that most of these suppressors would be in the CRISPRi machinery (e.g., dCas9, see Zhao et al., 2016, PMID: 27528508) and that the resulting strains would grow like wildtype. Do the final populations, diluted back to starting concentration, display the expected wildtype growth curve, or do they repeat the delayed lag phase, complicating the result?
 - d. While not required, attempting to delete the native spd1416/1417 genes in the presence of Zn⁺ (without gfp fusion backup) will rule out the possibility that Zn⁺ itself is suppressing the essential phenotype.

Additional comments

1. Lines 60-64: needs to be rewritten for clarity (or the passage should simply be removed).
2. Lines 95-97: Saying the absence of CRISPR/Cas is "likely because it interferes" with natural transformation would be better characterized as consistent with interference" or similar.
3. Figure 2C: Please consider whether this diagram is actually better than a simple Venn diagram. The atypical representation for overlap of datasets is of unclear value, and makes rapid

comprehension of the data more difficult.

4. Figure 3: There is not sufficient information in the legend describing the various phenotypes that are pointed out.

5. Lines 164-166: What was learned from the strains that could not be constructed? What characterized the failure mode? Please say more about this.

6. Lines 183-184: "we first analyzed the effects of CRISPRi-based repression on cell morphology using 69 genes that have been reported to be essential or crucial for normal pneumococcal growth" Needs reference.

7. The supplement (Fig. S2) contains examples of growth defect and lysis phenotypes, but as this is an important component of understanding the initial phenotype generation we would like to see more examples (in the supplement) and in particular several examples of what "no phenotype" and "both phenotypes" look like.

8. Figure 4D: *gatD* microscopy (bottom-left panel) is compelling on close inspection, but at current resolution, printed out to paper, the graphic does not make membrane localization very apparent.

Reviewer #3:

In this study Xue Liu et al. perform a CRISPRi assay using an arrayed library of 348 sgRNA targeting potentially essential genes of *S. pneumoniae* identified through a Tn-seq screen. The authors establish a practical CRISPRi screen in *S. pneumoniae* (note however that CRISPRi was already demonstrated in *S. pneumoniae* in PMID:23761437, this paper should be cited). By combining CRISPRi with high-content microscopy they are able to phenotype individual knockdowns and identify the function of several genes annotated as unknown. Overall this is a very nice study, the experiments are performed with all the appropriate controls and the paper is well written. One important point that I think is missing is a more detailed analysis of the targeted genes that do not show a growth phenotype. Why were these genes identified as essential by other methods and not CRISPRi?

Minor comments:

L215-225: this part is poorly written. It should be explained at the beginning of the paragraph that both C and N-terminal fusions were attempted and the nomenclature of the plasmids explained, otherwise the reader is lost when going over the first mention of Pzn-spd1416-gfp.

Figure 2: I find the representation of panel D a little confusing. The scale of the axis are different and the x-axis basically provides no information. Maybe showing a distribution plot would be easier to read?

Table 1 is poorly formatted and hard to understand

1st Revision - authors' response

03 March 2017

Authors replies to the referees.

We would like to thank the reviewers for their supportive comments and suggestions. We think that the changes and additional experiments we have made to address these points have considerably improved the manuscript. We provide a detailed point-by-point response and a list of the specific changes made to the Figures and Supporting Information below. The changes made in the text in line with comments from referees are highlighted.

The specific changes made to Figures and Supporting Information are as follows:

Main figures:

Figure 1: Minor formatting changes

Figure 2: Panel C was reformatted in line with the comment from Reviewer #2; Panel D

was reformatted and extended in line with the comment from Reviewer #3.

Figure 3: Minor formatting changes

Figure 4: Micrograph of GatD in panel D was replaced in line with the comment from Reviewer #2.

Figure 5: Minor formatting changes

Figure 6 was extended to include results of all the tested *clp* genes (Panel C) in line with the comment from Reviewer #1.

Extended view figures:

EV1: old Figure S1, a new panel B was added.

EV2 is the old Figure S2 with new panels D and E to show more examples of growth defects, in line with the comment from Reviewer #2.

EV3 is new and shows the presence of suppression mutations of the CRISPRi system in line with comment from Reviewer #2.

EV4: same as old Figure S15.

Appendix Figures:

Appendix Figure S1 is new and shows the growth curves of CRISPRi strains targeting dispensable genes (as negative controls), and examples of essential genes, in line with comments from Reviewer #1.

Appendix Figure S2 is new and demonstrates the polar effect of CRISPRi with RNA-Seq data and operons containing both essential and dispensable genes, as suggested by Reviewer #1 and Reviewer #2.

Appendix Figure S3-S14: as before

Appendix Figure S15 is new and shows the TA western blotting results with a control gene (*tarI*) in line with the comments from Reviewer #1.

Specific point-by-point responses to the referees' comments and other changes made to the manuscript are shown below.

Reviewer #1:

The study by Liu and coauthors is an elegant example of the adaptation of CRISPRi technology to the phenotypization of bacterial genes. This technology has previously been adapted to a few other bacteria, and the scale or the current work certainly merits significant attention. The authors focus their work on genes predicted to be essential by their transposon mutagenesis experiments, designing a library of sgRNAs that spans ~15% of the protein coding genes. From careful phenotypic analysis of several candidates the authors help elucidate genes involved in peptidoglycan and teichoic acid synthesis, and natural competence. The latter experiments are particularly striking, in part because of the quality of the negative controls provided by the non-participating subunits.

My enthusiasm for the study is mollified by the general lack of key controls present in the study. For instance, an adequate number of targets predicted to be dispensable in their transposon experiment could have served to assess the precision of their CRISPRi approach. Such controls could be easily included and would significantly improve the study, helping to assess the reliability of the CRISPRi method, and increasing the confidence of a few key observations. Provided with the adequate controls, I have no doubt the work will have a tremendous impact on the study of *Streptococcus pneumoniae*.

We thank the referee for the overall positive evaluation of our study. To alleviate the referee's major concern, we now included CRISPRi data of several non-essential genes as controls to assess the precision of the CRISPRi approach (new Appendix Figure S1). For details, see our responses to the specific comments below.

MAJOR COMMENTS:

Fig. 2. Given that polar effects have been well documented when using CRISPRi

in bacteria, the authors should analyze their screening results in the context of the operon architecture. When targeting the same operon, what is the agreement of different sgRNAs? Such an analysis would be facilitated by including selected sgRNAs against operons that contain both essential and dispensable genes. Are there any cases where the CRISPRi effect on a known dispensable operon, extends into an essential operon? Quantifying and displaying the magnitude of such effects, for instance by performing RNAseq experiments on a couple of other genes, would help determine the degree of confidence that readers should have on the authors' data set. However, a simple analysis of the existing data is probably sufficient to satisfy this point, since there are a lot of genes that did not show a phenotype in their analysis.

We agree that this is an interesting question and we have tackled this in several ways. First, we have reanalyzed our RNA-Seq data with the *luc* reporter strain and indeed observed the polar effects on downstream of the *luc* gene as we did not include a transcription terminator after the *luc* gene (new Appendix Figure S2A, Table EV2). This also explains the majority of significantly downregulated genes upon dCas9 induction (Fig. 1D). To test the polar effect with real operon examples, we selected 6 operons that contain both essential and dispensable genes, based on a preliminary study about identification of pneumococcal transcription start sites in our group (Slager et al., unpublished). As shown in new Appendix Figure S2, a polar effect could be observed upon dCas9 induction in all cases. Thus, whenever an essential gene was downstream of a non-essential gene, growth was still affected when the sgRNA targeted transcription of the dispensable gene. To validate that these targeted dispensable genes are truly non-essential, we made non-polar replacement mutants (Appendix Figure 2B and 2C). This validated the Tn-seq data and showed that the genes can be readily replaced by an antibiotic resistance marker (without promoter and without terminator). Interestingly, it seems that in some cases there can even be a polar effect on upstream genes (Appendix Fig S2D, operon number 5, *spd_1895*). However, when the operon contains multiple genes and the targeted dispensable gene localizes at the end of the operon, the polar effect on upstream genes is much less (Appendix Figure S2D, operon number 6, *lytC*). This new data is now described in the revised manuscript (Lines 190-191).

Since this study mainly focused on essential genes, we didn't have many CRISPRi strains targeting dispensable operons. In our study, we didn't observe any case where the CRISPRi effect on a known dispensable operon extends into an essential operon according to the growth analysis.

Fig. 2C. Were any of the genes predicted to be dispensable further analyzed? Presumably some of these were particularly implicated in essential functions by the transposon experiments, yet appeared dispensable by CRISPRi. Understanding the source of the discrepancy, could help establish the reliability of the methods and the ultimate relevance of the screening results.

Indeed, some genes were identified as essential in Tn-seq, but appeared dispensable by CRISPRi, and this discrepancy can be caused by several reasons. These include insufficient repression of transcription for instance because the sgRNA targets a PAM site far away from the TSS, some proteins are sufficient at just a few molecule numbers, poor access of the sgRNA-dCas9 complex to the target DNA, conditional essentiality (growth on plates vs. growth in liquid), or polar effects within the operon alleviating the essentiality. This is now clarified in the Discussion of the revised manuscript (Lines 384-398).

Fig. 3. Measuring growth curved is a central part of the study's phenotypic assessment, and would benefit from a set of genes pre-selected as controls. Is there a set of genes that are known to be dispensable and could be used as controls? This is particularly important given the documented polar effects of the CRISPRi system.

This is a good point, and we generated 8 new strains targeting non-essential genes. CRISPRi knockdown of these randomly selected 8 dispensable genes didn't significantly influence growth of *S. pneumoniae* (Appendix Figure S1A), main text on Lines 183-189.

Can sgRNAs against dispensable genes at the end of an operon be used to repress such genes without affecting the rest of the operon?

To test this, we included two operons with dispensable genes at the end, while with essential genes are upstream (Appendix Figure S2D, operon number 5 and number 6). Interestingly, in operon number 5, we observed that CRISPRi repression of dispensable gene *spd_1895* led to a significant growth defect, probably due to the polar effect on the upstream essential gene *gltX*. However, with operon number 6, which consists of 4 genes, it seems that the effect on upstream genes by CRISPRi repression of the dispensable gene *lytC* cannot be detected in the growth analysis.

Fig. 4. Depletion of the GFP-tagged proteins should be demonstrated, to confirm the effects of the genetic manipulations. This could be ideally performed by immunoblotting against the GFP portion of the fusion proteins, compared to a control. Alternatively the strains could be compared by FACS, provided an adequate counterstain to gate on the bacteria.

The zinc-inducible system we used in this study has been reported to be suitable for the fine control of *gfp*-fusion gene expression and for protein depletion experiments in *S. pneumoniae* (Eberhardt et. al., 2009 Molecular Microbiology), and has been applied for depletion in *S. pneumoniae* by several studies and confirmed as functional (Zapun, et. al., 2013 ACS Chem. Biol; Kjos, et. al., 2016 Plos Pathogen.; Beilharz et. al., 2012 PNAS). Here, in this study, we did in-gel fluorescence of GFP and proved that the GFP is successfully fused to the target protein (Appendix Fig S14), and we indeed observed that the PZn depletion strains responded to the inducer, Zn²⁺ (Figure 4A, 5A and Appendix Fig S13A). Nevertheless, we have performed a depletion assay as requested by the reviewer, and, although the resolution is not great, the image below clearly shows that the zinc-inducible system can be reliably used to deplete proteins.

Fig. 5E. I recognize that the monoclonal antibody against phosphorylcholine has previously been used; however, to my knowledge, the small species found in the TarP/Q knockdowns have not been documented. If I am wrong on this point, please reference such an observation appropriately. Alternatively, the authors should include a mutant of an upstream member of the pathway to fully identify the bands they claim to be repeating units, and thereby validate their assay.

Good point. The monoclonal antibody used to detect teichoic acids can only recognize molecules with choline modification, and in the teichoic acid biosynthesis pathway of *S. pneumoniae*, choline modification of precursors happens just before polymerization (Denapaite et al, 2012 Microbial Drug Resistance). So far, to the best of our knowledge, no gene has been implicated in the polymerization of TA precursors in *S. pneumoniae*, and thus there is no documentation of the small species observed in our gels. In the revised manuscript, we included a CRISPRi strain targeting *tarI* of the *licI* locus, which is involved in the very early step of precursor synthesis as a control, and repeated

the experiment (new Appendix Fig S15). Note that *tarI* is cotranscribed with the other 4 genes of the *licI* locus, including *tarJ*, *licA*, *licB* and *licC*. Theoretically, CRISPRi knockdown of *tarI* will lead to repression of the whole *licI* locus, and thus block the synthesis of TA precursors. In line with this, we observed reduction of teichoic acid chains when *tarI* is repressed by CRISPRi, different from the band pattern of repression of the studied *tarP* and *tarQ*. For text related with this point, see lines 337-343.

Fig. 6B. As mentioned above, I believe the strength of these experiments is the lack of an effect in the case of the *clpC*, *clpE* and *clpL* strains. Finding a way to summarize the results from the growth curves in a bar graph could be used to bring these important controls into the main text.

Many thanks for this suggestion. We now added a new panel C in Figure 6 summarizing the results of all the studied *clp* genes.

MINOR COMMENTS:

lines 222-224: The sentence needs clarification. Was a different construct than the one described above used? It is not clear from the context what is being introduced by erythromycin selection.

Yes, different from the above used N-terminal GFP fusions, we used a strain ectopically expressing C-terminal GFP fusions. For clarification, we have rewritten this paragraph, Lines 245-258.

line 266: Inability to replace the genes merely suggests essentiality; it does not "confirm" it.

Corrected. Line 301.

Reviewer #2:

Summary

Liu et. al. constructed a genome scale library of essential gene knockdown strains by using CRISPR interference (CRISPRi) in *S. pneumoniae*, and identified phenotypes for the knockdown strains using a systematic microscopy screen. The importance of this study is that it is the first CRISPRi screen in non-rod-shaped bacteria, as well as the first one in a pathogen, and will be a valuable resource for the research community. It further shows the utility of using CRISPRi as for studying essential gene function. The authors follow up on a number of phenotypes to ascribe function to essential genes without previous functional description. Several aspects of the manuscript should be clarified or improved, as described below.

General Comments

1. The overarching theme of this work is to ascribe function to poorly characterized genes that are essential to the organism. Therefore, it is incumbent on the authors to carefully describe both existing annotation and how the gene set to be analyzed was chosen.

a. which databases were used?1. The au

In this study, the annotation from NCBI (Version: CP000410.1, updated on 31-JAN-2015) was used. We have modified the text, and added this information in the introduction (Lines 78-79) and result sections (Lines 222-223; 237) of the revised manuscript.

b. Line 347: please expand on the source and nature of the mis-annotation.

The source has been added in the revised manuscript.

“Note that *spd_2030* (*dnaC*) was mis-annotated as *dnaB* in several databases, such as in NCBI (ProteinID:ABJ54728), KEGG (Entry: SPD_2030), UniProt (Entry: A0A0H2ZNF7), which may be due to the different naming of primosomal proteins in *E. coli* and *Bacillus subtilis* (Briggs et al, 2012; Smits et al, 2011)”, lines 403-406.

c. In general, Tn-seq screens overestimate the number of essential genes. What criteria did you use for deciding on the set of genes to profile?

The criteria for selection of genes targeted in our CRISPRi library are described in the Materials and Methods section (lines 472-480) and Table EV1 in detail. To select genes for our CRISPRi study, we used the output of three independent Tn-seq studies: our Tn-seq study on strain D39 and two published Tn-seq studies using strain TIGR4 (van Opijnen et al., 2009 Nat Methods; van Opijnen et al., 2012 Genome research). We first assigned a score to each gene based on the fitness cost obtained from each Tn-seq study, and then summarized the essentiality score of each gene of the 3 Tn-seq studies. Then the genes with relatively higher essentiality scores were selected (Table EV1). To clarify this, we have modified the main text of the revised manuscript (Lines 113-119). Moreover, to explain why Tn-seq screens overestimate the number of essential genes, we added a paragraph in the Discussion part (lines 386-398).

d. Lines 207-208: Please explain in the text how you determined which subset to phenotype.

To explore the function of unknown genes, we focused on a subset of the 348 CRISPRi knockdowns targeting genes that were annotated as “hypothetical protein” in the NCBI database (Version: CP000410.1, updated on 31-JAN-2015). This has now been clarified in the revised manuscript (Lines 235-238).

2. Sequence similarity between investigated genes and other genes should be better described in the body of the text, so that the reader can distinguish between cases where sequence similarity might have sufficed to establish function and cases where the authors follow-up was essential. At present, sequence similarity is discussed only in the supplement. For example in the investigation of dnaB/dnaD and spd1405/spd1522 the sequence similarities and bioinformatics likely would have sufficed to establish function, whereas for tarP/tarQ, the CRISPRi phenotype and follow-up was essential. Example: In Line 202: "our data revealed" should be clarified, at least in part by bringing supplemental data about sequence similarity into the body of the text.

Agreed and we have clarified this in the revised manuscript (Lines 227-232).

3. For the genes explored in detail, care should be taken to explicate the genomic context and enumerate the potential polar effects within operons. The recent work on the B. subtilis essential gene set showed that CRISPRi exhibits polarity both on downstream and upstream genes in an operon. Therefore, if the knockdown gene of interest is in an operon, the authors should mention the possibility that several genes in addition to the targeted gene may be responsible for the phenotype. One example of this issue is that the authors show the rpoA (alpha subunit of RNA polymerase) as an indicator of transcription. However, rpoA is in an operon with ribosomal proteins and the phenotype could be one resulting from those genes.

This is a good point also raised by Referee #1. To highlight the polar effects of CRISPRi technology, we made a new figure (Appendix Fig S2), see our responses to Referee 1. We have clarified this also in the text (Line 190-191).

4. Various experiments are missing important controls. Examples:

a. Figure 1D: please add a control comparing WT (+luc, no-dCas9) to the dCas9 strain (+dCas9, +luc) to verify that Cas9 itself is not responsible for changes.

b. Lines 149-150: Is the sequence targeted by sgRNA_{luc} also found in the com operon (as a perfect or imperfect match)? This might explain the slight-but reproducible -knockdown of com genes in Fig. 1D. It would be a good idea to mention here that you ruled out perfect or near-perfect matches.

We had included the proper controls but made this not clear enough. This has been corrected in the revised manuscript. In the RNA-Seq study, we included three different samples:

sample 1: WT(+luc, +sgRNA_{luc}, +dCas9) without IPTG

sample 2: WT(+luc, +sgRNA_{luc}, +dCas9) with 1 mM IPTG

sample 3: WT(+*luc*, -*sgRNA*_{luc}, +*dCas9*) with 1 mM IPTG.

This shows that expression of *dcas9* is stringently repressed by LacI in the absence of IPTG in Figure EV1, and RNA-Seq data of the sample 1 also confirmed this, because the RPKM-value of *dcas9* gene is practically zero in this sample. Based on these observations, the sample 1 essentially acts as a no-*dCas9* control. What's more, in the supplementary table 2, we provided the comparative analysis of sample 2 vs. sample 1; and sample 3 vs. sample 1. From the results, we can see that high expression of *dcas9* in the strain without *sgRNA* didn't cause any significant changes of the *luc* and the downstream genes, which was observed in the strain with *sgRNA*, confirming that the changes are not caused by *dCas9* itself but by the functional CRISPRi system which requires expression of both *dCas9* and *sgRNA*.

In addition, we observed a slight down-regulation of several competence-related genes in both sample 2 and sample 3. This is probably due to the fact that the applied experimental conditions allowed for natural competence to develop. Since the expression of the competence regulon is strongly regulated, but also highly time-dependent, small changes in their expression are almost inevitable. We have also checked for perfect or imperfect matches of the *luc* *sgRNA* to *com* genes and have excluded this. This observation supports the idea that the slight repression of *com* genes does not rely on the CRISPRi system (addressing question b). These results are now better explained in the revised manuscript (Line 145-155).

c. Lines 340-342: If you really are picking up suppressors during the growth of CRISPRi knockdown strains, I would expect that most of these suppressors would be in the CRISPRi machinery (e.g., *dCas9*, see Zhao et al., 2016, PMID: 27528508) and that the resulting strains would grow like wildtype. Do the final populations, diluted back to starting concentration, display the expected wildtype growth curve, or do they repeat the delayed lag phase, complicating the result?

To answer this question, we performed the suggested experiment. The results are similar to the expectation of this reviewer (new Figure EV3). We grew 6 of the CRISPRi strains targeting different essential genes in C+Y medium with 1 mM IPTG. After long-time incubation (15 hours), bacterial cells were streaked onto agar plate to purify single colonies. Single colonies were picked and used for a new round of growth analysis. The data shows that all the 6 purified 'suppressor' strains do not respond to IPTG anymore. Further sequencing analysis of the *sgRNA* and *Plac-dcas9* of these strains showed that most of the suppression mutations mapped to the coding sequence of *dcas9* (Figure EV3B). Description of this result has been added into the revised manuscript, please refer to lines 191-198.

d. While not required, attempting to delete the native *spd1416/1417* genes in the presence of Zn⁺ (without *gfp* fusion backup) will rule out the possibility that Zn⁺ itself is suppressing the essential phenotype.

Good point and we agree that this was not clearly described. In fact, we did this control for every essential gene confirmation in this study, and found that attempts to knockout these genes all failed in the wild-type background with Zn²⁺, ruling out the possibility of Zn²⁺ suppressing the essential phenotype. To clarify this, we have added this information in the revised manuscript (Line 251-253).

Additional comments

1. Lines 60-64: needs to be rewritten for clarity (or the passage should simply be removed).

Done (passage was removed).

2. Lines 95-97: Saying the absence of CRISPR/Cas is "likely because it interferes" with natural transformation would be better characterized as consistent with interference" or similar.

We have modified this as below:

"Note that *S. pneumoniae* does not contain an endogenous CRISPR/Cas system, consistent with interference with natural transformation and thereby lateral gene

transfer that is crucial for pneumococcal host adaptation (Bikard et al, 2012).”

3. Figure 2C: Please consider whether this diagram is actually better than a simple Venn diagram. The atypical representation for overlap of datasets is of unclear value, and makes rapid comprehension of the data more difficult.

Thanks for this advice. We have improved this figure, please refer to the new Figure 2C.

4. Figure 3: There is not sufficient information in the legend describing the various phenotypes that are pointed out.

Thanks for the advice. We have extended the description in the legend. Please refer to the revised figure legend of Fig 3.

5. Lines 164-166: What was learned from the strains that could not be constructed? What characterized the failure mode? Please say more about this.

The failed 43 sgRNA cloning may be caused by some technical reasons, for example, quality of the primers that we ordered; technical failure of the primer annealing; the limitation of cloning efficiency of the mentioned infusion-cloning, etc. Since we intend to apply the designed CRISPRi technique as a high-throughput tool for genome-scale study, we didn't do multiple attempts to clone these sgRNAs to exclude the possibility of technical reason caused failure. We thus cannot conclude on any common characteristics of the failed sgRNAs. We have clarified this in the revised manuscript (Lines 169-172).

6. Lines 183-184: "we first analyzed the effects of CRISPRi-based repression on cell morphology using 69 genes that have been reported to be essential or crucial for normal pneumococcal growth" Needs reference.

We have added references for this (van Opijnen et al, 2009; van Opijnen & Camilli, 2012), see lines 204-208 of the revised manuscript.

7. The supplement (Fig. S2) contains examples of growth defect and lysis phenotypes, but as this is an important component of understanding the initial phenotype generation we would like to see more examples (in the supplement) and in particular several examples of what "no phenotype" and "both phenotypes" look like.

We agree and have added additional examples, see the new Appendix Figure S2B-E.

8. Figure 4D: gatD microscopy (bottom-left panel) is compelling on close inspection, but at current resolution, printed out to paper, the graphic does not make membrane localization very apparent.

Agreed and we have improved this figure, see new Figure 4D.

Reviewer #3:

In this study Xue Liu et al. perform a CRISPRi assay using an arrayed library of 348 sgRNA targeting potentially essential genes of *S. pneumoniae* identified through a Tn-seq screen. The authors establish a practical CRISPRi screen in *S. pneumoniae* (note however that CRISPRi was already demonstrated in *S. pneumoniae* in PMID:23761437, this paper should be cited). By combining CRISPRi with high-content microscopy they are able to phenotype individual knockdowns and identify the function of several genes annotated as unknown. Overall this is a very nice study, the experiments are performed with all the appropriate controls and the paper is well written. One important point that I think is missing is a more detailed analysis of the targeted genes that do not show a growth phenotype. Why were these genes identified as essential by other methods and not CRISPRi?

We thank the reviewer and apologize for not citing the mentioned paper, this has now been fixed.

We appreciate the advice to provide a more detailed analysis of the targeted genes that do not show a growth phenotype (see also our reply to Referee 1). To explain this in

more detail, we added a paragraph in the Discussion (lines 384-398).

Minor comments:

L215-225: this part is poorly written. It should be explained at the beginning of the paragraph that both C and N-terminal fusions were attempted and the nomenclature of the plasmids explained, otherwise the reader is lost when going over the first mention of Pzn-spd1416-gfp.

Agreed and we have rewritten this paragraph (Line 245-258).

Figure2: I find the representation of panel D a little confusing. The scale of the axis are different and the x-axis basically provides no information. Maybe showing a distribution plot would be easier to read?

We made a new panel D, providing negative controls compared with the CRISPRi strains, to show the growth feature of the knockdown strains. Please refer to the new panel D of Figure 2.

Table 1 is poorly formatted and hard to understand

The Table 1 was re-formatted. Please refer to the new Table 1.

2nd Editorial Decision

03 April 2017

Thank you again for sending us your revised manuscript. We have now heard back from the two referees who were asked to evaluate your study. As you will see below they think that most issues have been satisfactorily addressed. However, reviewer #2 lists a few remaining concerns, which we would ask you to address in a minor revision.

In your revision, we would also like to ask you to address some remaining editorial issues listed below.

REFeree REVIEWS

Reviewer #1:

The authors have provided extensive new data, figures and clarifications that together answer all of my initial concerns. In my opinion, the paper has been greatly improved by these changes and will undoubtedly have a significant impact on the field.

Reviewer #2:

The authors revisions address most of our concerns, but there are twopoints that still need work.

1) "In general, Tn-seq screens overestimate the number of essential genes. What criteria did you use for deciding on the set of genes to profile?"

The presented table (EV1) and explanations in the new text are needlessly confusing. The provided "essentiality score" is not justified or characterized. There is no obvious meaning for "core genes", much less any reason to think that a 0.2 or a 0.5 is a good numerical score for a given characterization. Furthermore the multiple cutoff thresholds for sum-of-scores across different data subsets are not justified and appear to be internally inconsistent.

We suggest that something much much simpler would serve the authors far better. Something like "We chose to include all genes that we found to be essential, and added in genes that were found to be essential by these previous Tn-Seq studies", with any exceptions noted and briefly explained. The tables are useful for comparison, but the quasi-quantitative scores and thresholds are not.

2) "For the genes explored in detail, care should be taken to explicate the genomic context and enumerate the potential polar effects within operons. The recent work on the *B. subtilis* essential gene set showed that CRISPRi exhibits polarity both on downstream and upstream genes in an operon. Therefore, if the knockdown gene of interest is in an operon, the authors should mention the possibility that several genes in addition to the targeted gene may be responsible for the phenotype. One example of this issue is that the authors show the *rpoA* (alpha subunit of RNA polymerase) as an indicator of transcription. However, *rpoA* is in an operon with ribosomal proteins and the phenotype could be one resulting from those genes."

rpoA is still the presented exemplar for "Transcription" phenotype in Figure 3, despite the polar interactions with ribosomal genes. A different example should be used that does not share this complication (e.g. *rpoB*, *rpoC*, from figure S4).

It is still not clear whether the phenotypes presented in figure 3 are general to the annotation class, or specific to the presented gene. The text should address this question. The phenotypes presented in supplemental figures S3-S10 need explanations of observed phenotypes as have now been added to figure 3 to help clarify this point further. Ideally we would see representative exemplars of each phenotype in figure 3, reinforced by the other examples in S3-S10, to support the claim in 220-221 that this approach was useful for 'functional verification' of uncharacterized genes.

2nd Revision - authors' response

12 April 2017

We have made the following changes in the second revised version:

- 1) Changed the descriptions of selection of genes to be involved in the CRISPRi library as suggested by the reviewer #2;
- 2) Replaced the *rpoA* result with *rpoC*, which is cotranscribed with another transcription related gene *rpoB*, in Figure 3, as suggested by reviewer #2;
- 3) Explained the observed phenotype in Figure 3 as suggested by reviewer #2;

Corresponding Author Name: Jan-Willem Veening

Journal Submitted to: Molecular systems biology

Manuscript Number: MSB-16-7449